# Phthalocyanines Conjugated with Small Biologically Active Compounds for the Advanced Photodynamic Therapy: A Review

**DOI:** 10.3390/molecules30153297

**Published:** 2025-08-06

**Authors:** Kyrylo Chornovolenko, Tomasz Koczorowski

**Affiliations:** 1Chair and Department of Chemical Technology of Drugs, Poznan University of Medical Sciences, Rokietnicka 3, 60-806 Poznan, Poland; kchornovolenko@ump.edu.pl; 2Doctoral School, Poznan University of Medical Sciences, Bukowska 70, 60-812 Poznan, Poland

**Keywords:** photodynamic therapy, photosensitizer, phthalocyanine, conjugates, targeted therapy

## Abstract

Phthalocyanines (Pcs) are well-established photosensitizers in photodynamic therapy, valued for their strong light absorption, high singlet oxygen generation, and photostability. Recent advances have focused on covalently conjugating Pcs, particularly zinc phthalocyanines (ZnPcs), with a wide range of small bioactive molecules to improve selectivity, efficacy, and multifunctionality. These conjugates combine light-activated reactive oxygen species (ROS) production with targeted delivery and controlled release, offering enhanced treatment precision and reduced off-target toxicity. Chemotherapeutic agent conjugates, including those with erlotinib, doxorubicin, tamoxifen, and camptothecin, demonstrate receptor-mediated uptake, pH-responsive release, and synergistic anticancer effects, even overcoming multidrug resistance. Beyond oncology, ZnPc conjugates with antibiotics, anti-inflammatory drugs, antiparasitics, and antidepressants extend photodynamic therapy’s scope to antimicrobial and site-specific therapies. Targeting moieties such as folic acid, biotin, arginylglycylaspartic acid (RGD) and epidermal growth factor (EGF) peptides, carbohydrates, and amino acids have been employed to exploit overexpressed receptors in tumors, enhancing cellular uptake and tumor accumulation. Fluorescent dye and porphyrinoid conjugates further enrich these systems by enabling imaging-guided therapy, efficient energy transfer, and dual-mode activation through pH or enzyme-sensitive linkers. Despite these promising strategies, key challenges remain, including aggregation-induced quenching, poor aqueous solubility, synthetic complexity, and interference with ROS generation. In this review, the examples of Pc-based conjugates were described with particular interest on the synthetic procedures and optical properties of targeted compounds.

## 1. Introduction

Photodynamic therapy (PDT) has emerged as a clinically established treatment modality for a range of diseases, including cancer, precancerous lesions, and various skin and vascular conditions [1,2]. The mechanism of PDT involves the combined action of three essential components: a photosensitizer (PS), light of a specific wavelength, and molecular oxygen [3]. Upon light activation, the photosensitizer generates singlet oxygen and other reactive oxygen species (ROS), which are responsible for inducing localized cellular damage and death (Figure 1). The production of ROS by activated photosensitizers is not only used in anti-cancer treatment but also in photoinactivation of bacteria, fungi, and viruses, where the treatment method is called photodynamic antimicrobial chemotherapy (PACT) [4].

Compared to conventional treatment strategies such as surgery, chemotherapy, and radiotherapy, PDT offers several notable advantages [5]. First, photosensitizers remain inactive until exposed to light, thereby reducing systemic toxicity. Second, PDT exhibits a broad range of potential applications due to the non-specific oxidative damage caused by ROS, which not only kills tumor cells directly but may also damage tumor vasculature and stimulate an antitumor immune response. Third, the technique is minimally or noninvasive and highly selective [6]. Additionally, PDT can be effectively combined with other modalities such as chemotherapy, surgery, or radiation therapy and has been explored as a method to overcome multidrug resistance in both cancer and bacterial infections. Finally, the integration of PSs with nanomaterials has enhanced their biomedical applications, particularly through mechanisms such as the enhanced permeability and retention (EPR) effect [7].

A critical determinant of PDT efficacy is the nature and performance of the photosensitizer [8]. Classically defined as molecules that, upon absorbing light, induce chemical or physical changes in surrounding biomolecules, photosensitizers must meet stringent criteria: strong absorption in the red or near-infrared (NIR) spectrum to facilitate deep tissue penetration, high phototoxicity under light exposure with minimal dark toxicity, stability and solubility in physiological conditions, preferential accumulation in the neoplastic tissue, and appropriate clearance profiles [9,10]. Furthermore, ideal photosensitizers for PDT should exhibit high triplet-state quantum yield (Φ_T_) and singlet oxygen quantum yield (Φ_Δ_) to maximize ROS generation [11]. A principal limitation of many current photosensitizers is their suboptimal absorption in the therapeutic optical window (600–800 nm), where tissue penetration is maximized and where the interference from endogenous chromophores such as hemoglobin is minimized. Beyond 850–900 nm, photons lack sufficient energy to initiate photochemical reactions, underscoring the need for photosensitizers that operate efficiently within this therapeutic window [12].

Among second-generation photosensitizers, phthalocyanines (Pcs) have shown exceptional promise [13]. First synthesized in 1907, Pcs have since found widespread use in industrial applications due to their robust photophysical properties. These macrocyclic compounds, structurally related to porphyrins, exhibit intense absorption in the far-red region (~670 nm), which is advantageous for tissue penetration. Their optical properties can be tuned via peripheral substituents or metal coordination, making them highly attractive for PDT applications [14]. Several Pc derivatives have already been approved for clinical use in oncology and ophthalmology [2]. Unsubstituted zinc(II) phthalocyanine (ZnPc) is one of the most extensively studied Pc-based photosensitizers, recognized for its high PDT efficacy [15]. However, its clinical utility is hindered by poor water solubility, necessitating formulation strategies (e.g., liposomes, surfactants) or chemical modifications to improve bioavailability and therapeutic performance [16]. Another major challenge in conventional PDT is the limited selectivity of photosensitizers for cancerous over healthy tissues. An optimal delivery system should achieve selective accumulation in tumor cells while minimizing off-target effects. Enhancing the oncotropicity or tumor-targeting ability of photosensitizers is a key strategy for improving therapeutic outcomes and reducing systemic toxicity [17,18]. To date, many Pcs in clinical trials rely on passive targeting via the EPR effect. This phenomenon exploits the leaky vasculature and poor lymphatic drainage of tumors to enable the selective accumulation of large molecules. However, recent findings have questioned the translational reliability of EPR-based targeting, especially in moving from preclinical animal models to human clinical settings [11]. Therefore, active-targeting strategies involving the conjugation of tumor-specific ligands to photosensitizers are increasingly being explored as potential more effective alternatives [19].

Naturally derived targeting moieties such as peptides, antibodies, aptamers, and lectins have shown promise in enhancing the cellular uptake and tumor specificity of photosensitizers [20,21]. However, challenges related to synthetic complexity, in vivo stability, and potential immunogenicity remain. In response, small molecule-based targeting agents, including PEG derivatives with low molecular weight, short peptides, and vitamins such as folic acid, riboflavin, vitamin B_12_, and biotin, are being investigated for their ability to improve tumor selectivity and cellular uptake without provoking significant immune responses [22,23,24]. Up to this date, numerous strategies have been utilized for the synthesis of covalent conjugates between phthalocyanines and bioactive molecules, usually providing targeted therapy or enhancing bioavailability of a photosensitizer, as we demonstrated in this article.

In this review, we would like to summarize the recent data regarding the synthesis and optical properties of covalent conjugates of phthalocyanines as second-generation efficient photosensitizers and diverse organic compounds with biological activity, starting from common drugs, through targeting moieties, ending up with small molecules such as amino acids, chalcones, or biotin. The conjugates obtained by axial ligation of organic compounds to the central metal cation of the macrocyclic core were deliberately omitted as they are based on coordination bonds. Our approach relies on the comparison of reaction types used for conjugate formation, presented as a separate chapter, and limitations of targeted compounds influencing overall outcomes related to their photoactivity and photocytotoxicity.

## 2. Synthesis of Phthalocyanines

Phthalocyanines can be synthesized through several methods, with one of the most prominent being the Linstead macrocyclization (Figure 1), particularly effective for producing magnesium phthalocyanine derivatives [25]. In this approach, the cyclotetramerization of phthalonitrile or phthalic anhydride derivatives is carried out in the presence of magnesium butanolate, which acts both as a magnesium source and a catalyst. Notably, magnesium butanolate is often generated in situ by reacting magnesium metal with 1-butanol under reflux conditions [26]. Once formed, the magnesium butanolate facilitates the condensation of four phthalonitrile units to form the macrocyclic phthalocyanine core coordinated to a central magnesium ion. Instead of 1-butanol, successful attempts were performed using ethanol [27].

The other common approaches rely on the thermal cyclization of phthalonitrile with a metal salt (such as copper, cobalt, or zinc salts) under high temperatures (usually above 200 °C) in high-boiling solvents [28]. These reactions lead to metal phthalocyanines with high purity and relatively good yields. Another widely used method involves the reaction of phthalic anhydride with urea and a metal salt in the presence of a basic catalyst such as ammonium molybdate or ammonium heptamolybdate [29]. This process typically requires heating and is carried out under nitrogen to prevent oxidation. The last approach, the solventless melt method, usually performed in microwave reactors [30], entails heating the starting materials directly without a solvent, which can be more environmentally friendly and cost-effective.

Mixed macrocyclizations, particularly the 3 + 1 approach (Figure 2), are a powerful method for synthesizing unsymmetrical phthalocyanine derivatives. In this strategy, three identical phthalonitrile units (usually 1,2-dicyanobenzene) are reacted with a fourth, structurally distinct phthalonitrile under controlled cyclotetramerization conditions (Figure 2) [31]. The 3 + 1 approach enables the incorporation of a single, different substituent or functional group into the phthalocyanine ring, breaking the symmetry of the macrocycle [31]. This unsymmetrical structure can significantly alter the electronic, optical, and solubility properties of the resulting compound. Achieving good selectivity in 3 + 1 cyclizations often requires careful control of reaction stoichiometry, temperature, and sometimes the use of templating metal ions. Otherwise, it leads to a statistical mixture of both symmetrical and unsymmetrical derivatives [32].

## 3. Strategies for Conjugate Synthesis

Based on the articles reviewed in this study, one can distinguish different strategies employed for the covalent conjugation of biologically active compounds to the phthalocyanine macrocyclic ligand. The first differentiation can rely on the formation of covalent bonding between a previously synthesized phthalocyanine derivative and the targeting compound or the initial conjugation of a biomolecule to a nitrile precursor with subsequent cyclotetramerization reaction. The second distinction embraces direct attachment to either Pc or nitrile derivative or indirect conjugation via aromatic or alkyl linker.

The synthetic approaches leading to the functionalization of a nitrile precursor for further macrocyclization include the nitro-group displacement reaction of a hydroxyl-terminated PEG linker to 3- or 4-nitrophthalonitrile.

The conjugation of biomolecules to previously-obtained phthalocyanine employed reactions such as (Figure 2) (i) the “click chemistry” approach – a triazole formation by copper-catalyzed azide-alkyl cycloaddition (CuAAC); (ii) amide bond formation between carboxyl-terminated Pc and proper alcohol (usually catalyzed by carbodiimide derivatives like EDC or DCC); (iii) Sonogashira cross-coupling with C≡C bond formation; (iv) Buchwald–Hartwig coupling; (v) hydrazone bond formation; (vi) transesterification reaction; (vii) N-acylation reaction (amide bond formation between acyl chloride and amine); (viii) sulfamidation; (ix) N-alkylation; and (x) esterification.

Several types of linkers have been used for the special separation of a phthalocyanine moiety and a biologically active small compound (Table 1). The most pronounced are ethylene glycols in their oligomeric (OEG) or polymeric (PEG) forms, widely utilized thanks to their biocompatibility and solubilizing properties. Amino acids like tyramine and even peptides (both cleavable and non-cleavable) are also employed. pH-sensitive linkers such as hydrazone or ROS-cleavable thioketal linkers are used wherever there is a need to trigger the compound’s release in a controlled way (Figure 3).

Triazole linkers formed in the copper-catalyzed cycloaddition between alkyl derivatives and azide-terminated compounds are widely used as they represent the “click chemistry” approach of obtaining new conjugates quickly in mild conditions. Among the others, alkyl chains and aryl-alkyl linkers are employed to increase the lipophilicity of the obtained conjugate. Last but not least, in a few cases, such compounds as thiourea, aminoacrylate, or cyclopentadiene have been used to combine a phthalocyanine ring and a biomolecule.

In the following chapters and subchapters, examples of phthalocyanine–biomolecule conjugates are briefly described alongside illustrations of the chemical structures of the targeted compounds. The conjugates are grouped according to their functional role, such as therapeutic agents, targeting moieties, or imaging probes, to provide a clearer overview of their relevance in advanced photodynamic therapy.

## 4. Phthalocyanines Conjugated with Therapeutic Agents

### 4.1. Anticancer Agents

Combining PDT with cytotoxic drugs has emerged as a powerful strategy to increase tumor selectivity and therapeutic efficiency [11]. In particular, the use of phthalocyanines as photosensitizers conjugated to small-molecule anticancer drugs enables dual-action agents capable of both light-activated cytotoxicity and receptor-mediated cellular targeting. Among these, epidermal growth factor receptor (EGFR) inhibitors like erlotinib have shown promise for treating cancers such as non-small cell lung and pancreatic carcinomas [33]. A pioneering study by Zhang et al. introduced the first erlotinib–ZnPc conjugates, designed to selectively target EGFR-overexpressing tumors [34]. In this work, two ZnPc derivatives (Figure 4, **I**) were synthesized using a modular strategy: erlotinib was attached through a 1,3-dipolar cycloaddition click reaction to oligoethylene glycol (OEG) linkers, followed by substitution at the α-position of phthalonitrile intermediates, and finalized via a statistical “3 + 1” cyclotetramerization. The resulting unsymmetrical ZnPcs had either three- or five-unit OEG spacers, which enhanced amphiphilicity and biocompatibility. Photophysically, both conjugates dissolved in DMF retained the sharp Q-band absorption at 678 nm and exhibited fluorescence quantum yields (Φ_F_ ≈ 0.27) nearly identical to the parent ZnPc (Φ_F_ = 0.26). Notably, the singlet oxygen quantum yields (Φ_Δ_) were 0.66 and 0.57 in DMF for the shorter and longer linkers, respectively, comparable or superior to the parent compound (Φ_Δ_ = 0.63). Although the conjugates lacked specific organelle localization, they achieved significant photodynamic efficacy against HepG2 liver cancer cells, with IC_50_ values ranging from 0.01 to 0.04 mM (λ = 670 nm, 1.5 J/cm^2^). The improved tumor affinity in vivo was attributed to EGFR-mediated cellular uptake.

Expanding on this concept, the same research group later synthesized a broader library of erlotinib–ZnPc conjugates with varying lengths of OEG chains (0–4 units) and different substitution patterns at the α and β positions [35]. As before, a three-step synthetic approach was employed, starting with a 1,3-dipolar cycloaddition to install erlotinib, followed by nucleophilic substitution of nitrophthalonitrile to obtain functionalized intermediates, and concluding with “3 + 1” macrocyclization. The resulting conjugates in DMF exhibited red-shifted Q-band absorption at 672–678 nm and fluorescence maxima at 680–686 nm. Photodynamically, they showed Φ_Δ_ values between 0.50 and 0.66 in DMF, with α-substituted derivatives generally outperforming their β counterparts. Despite the structural modifications, fluorescence yields remained stable (Φ_F_ ≈ 0.26–0.27). Biologically, the compounds demonstrated negligible dark cytotoxicity (up to 50 μM), but upon 670 nm light exposure (1.5 J/cm^2^), IC_50_ values dropped as low as 9.61 nM. EGFR-targeted uptake was confirmed via selective internalization in HepG2 over HELF cells, and subcellular tracking revealed lysosomal accumulation. Longer OEG chains led to modestly decreased uptake and phototoxicity, suggesting that hydrophilic bulk may hinder cell entry. Still, the compounds retained robust ROS generation and high PDT efficacy.

In contrast to the triazole-linked systems, Chen et al. [36] developed erlotinib–ZnPc conjugates using N-alkylation strategies, bypassing triazole formation altogether. OEG units (n = 1–3) were mono-tosylated and coupled directly to erlotinib, followed by substitution with nitrophthalonitrile. Interestingly, the macrocyclization process resulted in the loss of the 3-ethynylaniline motif from erlotinib, confirmed by NMR, IR, and HRMS. The final α- and β-substituted phthalocyanines in DMF showed Q-band absorptions at ~678 nm and ~673 nm, respectively. Despite these structural differences, the Φ_Δ_ values were largely unaffected by linker length or substitution position. All variants maintained photostability and ROS-generating capacity. Notably, EGFR-overexpressing HepG2 cells selectively internalized the conjugates over HELF cells. IC_50_ values following irradiation with 670 nm light ranged from 3.7 to 16.7 nM, with α-substituted derivatives displaying marginally better activity, further reinforcing the impact of substitution geometry on PDT efficacy.

A further advancement came with the development of ZnPc conjugates bearing multiple types of targeting ligands, including erlotinib, DAA1106, and PK11195—the latter two being known ligands of the translocator protein (TSPO) [37]. These six conjugates (Figure 4, **II**) were synthesized via a combination of O-/N-alkylation (for TSPO ligands) and dipolar cycloaddition (for erlotinib). Q-band red-shifts were observed, with absorption maxima in CHCl3 at 684–685 nm (TSPO conjugates) and up to 694 nm (erlotinib conjugates). While all compounds exhibited strong light absorption, biological activity diverged significantly: mono-substituted conjugates demonstrated potent photodynamic effects under 650 nm irradiation, while tetra-substituted analogs lacked phototoxicity, likely due to decreased cellular uptake and increased steric hindrance. Mitochondrial localization of the active conjugates was confirmed by fluorescence imaging. All retained low dark toxicity (IC_50_ ≥ 50 μM), ensuring safety under non-irradiated conditions.

To address multidrug resistance (MDR), Wei et al. designed ZnPc–lenvatinib conjugates where ZnPc and the VEGFR inhibitor were linked via alkyl chains of varying lengths (n = 0, 8, 12) [38]. Among them, ZnPc–C_8_–Len (Figure 4, **III**) emerged as the most effective, generating robust ROS and inducing oxidative stress via intracellular glutathione (GSH) depletion. This stress cascade led to Bcl-2 downregulation, caspase-3 activation, and notably, P-glycoprotein (P-gp) suppression—a hallmark of MDR reversal. PEG2000-PLA2000 nanoparticle encapsulation improved solubility and tumor targeting. The conjugate in DMSO solution retained Q-band absorption at 681 nm and emission at 687 nm, alongside high phototoxicity. Biological assays revealed 6.9-fold lower IC_50_ values under 660–670 nm light irradiation against MDR MCF7/ADR cells compared to ZnPc or Len alone. Although Len’s activity was slightly diminished in dark conditions, light-induced synergy between PDT and chemotherapy made this conjugate a compelling anti-MDR platform [16].

In another example of dual-mode therapy, ganetespib (Gan), an Hsp90 inhibitor [39], was conjugated to ZnPc to exploit the overexpression of extracellular Hsp90 in tumors [40]. The resulting Gan–ZnPc conjugate (Figure 4, **IV**), synthesized via sulfonation with an aliphatic linker, retained favorable photophysical characteristics. The Q-band red-shifted from 672 nm to 678 nm in DMSO, consistent with iodide substitution effects. Gan–ZnPc effectively produced ROS upon irradiation and displayed significant cytotoxicity even in the absence of light, demonstrating therapeutic efficacy through Hsp90 inhibition. In MCF7 cells, viability dropped below 20% at 100 nM after 24 h under red-light 660–670 nm. Fluorescence imaging showed preferential accumulation in tumor cells over healthy ones. The conjugate remained monomeric in fetal bovine serum, ensuring bioavailability and PDT activity [40].

ZnPc conjugates with classical chemotherapeutics have also shown synergistic promise. In one study, oxaliplatin was coupled to ZnPc via a triethylene glycol spacer (Figure 4, **V**) [41]. The synthesis began with alkylation of diethyl methylmalonate, followed by nucleophilic aromatic substitution with nitrophthalonitrile, macrocyclization, hydrolysis, and complexation with a platinum(II) unit. The conjugate retained a 672 nm Q-band in DMF and unaltered ROS generation. Importantly, it exhibited dual cytotoxicity—ROS-mediated phototoxicity and dark cytostatic action from the oxaliplatin moiety—achieving an IC_50_ of 0.11 µM in the presence of 610 nm light against HT29 cells, outperforming its non-platinum analog by 5-fold.

In the case of another classic chemotherapeutic, doxorubicin (Dox) was integrated into a ZnPc prodrug for tumor-specific activation. In the study by Ke et al., a peptide linker (Thr-Ser-Gly-Pro) responsive to the fibroblast activation protein (FAP) was employed to link Dox to monocarboxy-substituted ZnPc, forming the ZnPc–Dox prodrug (Figure 4, **VI**) [42]. The conjugate, recorded in DMF, had a Q-band at 674–676 nm, although ROS generation (Φ_Δ_ = 0.3–0.4) was initially quenched by Dox. Upon cleavage by FAP, overexpressed in cancer-associated fibroblasts, the active agents were released, restoring singlet oxygen production and cytotoxicity. In vitro studies confirmed selective killing of HepG2 cells only in the presence of FAP. In vivo, fluorescence imaging and therapeutic evaluation in tumor-bearing mice validated targeted prodrug activation, reducing systemic toxicity while enhancing tumor-specific PDT and chemotherapy efficacy [42].

Another notable advancement in the development of dual-functional chemo-photodynamic systems was reported by Wong et al., who engineered a sophisticated ZnPc–Dox conjugate-based nanoplatform aimed at achieving controlled, intracellular drug release and synergistic cytotoxicity through the integration of chemotherapy and PDT. The Wong group’s approach relies on the covalent linkage between photosensitizing zinc phthalocyanine and the anticancer drug Dox via a pH-sensitive hydrazone linker. This ZnPc–Dox conjugate was then encapsulated into mesoporous silica nanoparticles (MSNs) that were surface-functionalized with alkyne groups, enabling further chemical modification. The unique architecture of MSNs, characterized by high surface area, tunable porosity, and biocompatibility, was exploited to facilitate drug encapsulation, stability, and responsive release in acidic intracellular environments typical of tumor cells [43]. The conjugation methodology employed copper(I)-catalyzed azide–alkyne cycloaddition, a classic “click chemistry” reaction, to anchor the conjugates firmly within the nanochannels of MSNs, thereby mitigating premature release and enhancing formulation stability. To form the ZnPc–Dox conjugate, the peripheral ester group on the phthalocyanine ring was first reacted with an excess of hydrazine monohydrate to generate a hydrazide intermediate. This intermediate then underwent acid-catalyzed condensation with free base Dox, yielding the hydrazone-linked conjugate (Figure 4, **VII**) [43]. Upon cellular uptake by HepG2 liver cancer cells, both Dox and ZnPc fluorescence signals were detectable, confirming successful internalization and intracellular release of Dox under acidic conditions. Structural characterization and spectroscopic analyses demonstrated that the Q-band absorption at 672 nm in DMF, indicative of the ZnPc photophysical integrity, was retained post-conjugation. Despite a modest decrease in solubility due to Dox attachment, the conjugate maintained effective singlet oxygen production upon irradiation, a critical factor for PDT efficacy. The MSNs exhibited a hydrodynamic diameter of approximately 200 nm, and ZnPc loading ranged between 8.6 and 11.0 nmol/mg, reflecting efficient encapsulation. In vitro cytotoxicity assessments revealed a pronounced therapeutic benefit: light-activated (λ > 610 nm) ZnPc–Dox conjugates showed IC_50_ values between 0.18 and 0.25 μM, markedly superior to ZnPc alone (IC_50_ ~0.64 μM), highlighting the synergistic effects of combined photodynamic and chemotherapeutic actions. While some challenges, such as signal broadening and aggregation-induced quenching, were noted during synthesis, the overall design provided a robust platform for pH-sensitive, dual-modality cancer therapy [43].

Building on similar principles, Peng et al. developed a targeted drug delivery system by synthesizing a suite of ZnPc–Dox conjugates that exploit fibroblast activation protein (FAP) for site-specific release. The photosensitizer used was a carboxyl mono-substituted ZnPc, to which Dox was attached via peptide linkers. Two categories of linkers were explored: FAP-cleavable Gly-Pro dipeptide linkers and non-cleavable analogs, allowing for comparison of enzymatic responsiveness (Figure 4, **VIII**) [44]. The conjugation process involved classical amide bond-forming techniques using EDCI/NHS or HOBt activation strategies. Due to the steric and solubility challenges inherent in such multifunctional constructs, protective group chemistry and polar solvents were employed to improve reaction yields and product stability. Optical characterization revealed that the conjugates in DMF solution retained Q-band absorptions in the range of 672–682 nm and emitted in the 683–692 nm window. However, fluorescence quantum yields were suppressed relative to native ZnPc (0.01–0.25 vs 0.28), likely due to photoinduced electron transfer (PET) effects from the conjugated Dox moiety. Similarly, singlet oxygen generation was diminished (Φ_Δ_ ranging from 0.09 to 0.69, compared to 0.56 for the parent ZnPc), indicative of trade-offs between photodynamic function and chemotherapeutic payload integration. Among the tested conjugates, Pc–Gly–Pro–Dox displayed the most efficient FAP-mediated cleavage and release of the Dox payload, leading to enhanced tumor-specific fluorescence recovery and cytotoxicity [44].

In a parallel effort targeting hormone-responsive cancers, Zhang et al. synthesized a tamoxifen–ZnPc conjugate to integrate PDT with estrogen receptor (ER)-targeted endocrine therapy. The synthetic strategy commenced with the N-demethylation of tamoxifen to expose a reactive amine group, which was subsequently modified with a triethylene glycol spacer via nucleophilic substitution. This intermediate was then coupled to 3-nitrophthalonitrile to form a functionalized phthalonitrile precursor, which underwent mixed cyclotetramerization with unsubstituted phthalonitrile in the presence of Zn(OAc)_2_ and DBU to yield an unsymmetrical ZnPc conjugate in 9% yield (Figure 4, **IX**) [45]. The amphiphilic conjugate maintained water solubility and photodynamic functionality, as evidenced by its sharp Q-band absorption in DMF at 678 nm and emission at 688 nm. A singlet oxygen quantum yield of Φ_Δ_ = 0.63 in DMF confirmed its potential for PDT, despite a reduced fluorescence quantum yield (Φ_F_ = 0.17) relative to the parent ZnPc. Functional assays revealed pronounced ER-targeted behavior: the uptake and phototoxicity of the conjugate in MCF-7 cells (ER-positive) were significantly diminished upon competition with 17β-estradiol, validating receptor-mediated endocytosis. Confocal imaging localized the conjugate primarily in lysosomes. The drug exhibited ~50% dark cytotoxicity at 12.5 μM due to the tamoxifen unit while maintaining high photocytotoxic efficacy (λ = 670 nm), unlike the parent ZnPc [45].

Subsequently, Zhang and colleagues expanded on this approach by synthesizing a series of α- and β-substituted tamoxifen–ZnPc conjugates through a refined four-step procedure: tamoxifen demethylation, OEG spacer attachment, coupling to nitrophthalonitriles, and final cyclotetramerization with Zn(II) salts [46]. This yielded conjugates in moderate yields (7–14%), with the substitution position significantly influencing photophysical outcomes. α-Substituted analogs in DMF absorbed at 678 nm and emitted at ~686 nm, while β-substituted derivatives showed slightly blue-shifted spectra. All conjugates demonstrated robust singlet oxygen production (Φ_Δ_ = 0.51–0.63), with α-substituted versions outperforming in both phototoxicity under light irradiation λ = 670 nm (IC_50_ = 13.80–89.54 nM) and dark cytotoxicity, confirming superior therapeutic profiles for this subclass [46].

In a distinct chemotherapeutic coupling strategy, Peng et al. investigated chlorambucil (CLB) as the cytotoxic partner in ZnPc and SiPc conjugates. A mono-α-substituted ZnPc and an axially di-substituted silicon(IV) phthalocyanine (SiPc) were synthesized and functionalized with Boc-protected tyramine spacers, enabling EDCI/HOBt-mediated amide coupling with CLB (Figure 4, **X**) [47]. Although the parent phthalocyanines displayed potent PDT activity against HepG2 cells irradiated with a red light λ > 610 nm (IC50 values: 31 nM for ZnPc, 9 nM for SiPc), conjugation to CLB drastically reduced their efficacy (IC_50_: 0.20 mM and 17.47 mM, respectively). This decline correlated with diminished fluorescence and lower singlet oxygen quantum yields, underscoring that hydrophobicity and steric hindrance can severely impact intracellular uptake and ROS generation, leading to loss of therapeutic synergy [47].

Finally, in a comprehensive platform integrating multiple therapies, Martínez-Edó et al. developed a multifunctional MSN-based nanocarrier comprising three therapeutic components. The system encapsulated a ZnPc–camptothecin (CPT) conjugate via a cleavable amphiphilic PEG linker and separately anchored Dox on the particle surface using a dihydrazide–PEG linker (Figure 4, **XI**) [48]. The ZnPc–CPT conjugate was synthesized through sequential steps involving PEGylation, deprotection, and amide bond formation. Despite moderate yields (~35%), the resultant conjugate retained Q-band absorption at 679 nm and efficient singlet oxygen generation. The inclusion of PEG linkers improved solubility and cytosolic compatibility. Upon cellular uptake, fluorescence imaging showed cytoplasmic localization of ZnPc, with Dox and CPT localizing in both the cytoplasm and nucleus. Light irradiation (λ > 610 nm, 14 J/cm^2^, 10 min) significantly increased cytotoxicity via PDT and chemotherapy synergy. Notably, combination index (CI) analysis in HeLa cells yielded a value of 0.46, indicating strong synergistic effects, which resulted in approximately 80% cancer cell death across both HeLa and HepG2 models. This multi-component platform demonstrates the promise of integrating spatial and temporal control mechanisms for optimized cancer therapy [48].

In summary, the integration of chemotherapeutic moieties such as oxaliplatin [41], doxorubicin [42], erlotinib [36], and ganetespib [40], or the strategic attachment of targeting ligands [37], enables synergistic anti-cancer effects through dual or conditional activation mechanisms. Key findings demonstrate that such conjugates maintain or improve photophysical properties while reducing off-target effects, achieving enhanced intracellular ROS generation, selective tumor accumulation, and potent cytotoxicity upon irradiation. Crucially, structural features such as linker type, substitution position, and ligand valency significantly influence cell penetration and therapeutic performance, providing valuable insights for the future development of smart, tumor-specific PDT agents [36,37,40,41,42].

### 4.2. Other Therapeutic Drugs

In the 21st century, phthalocyanines have been covalently conjugated with drugs from different ATC groups, not only belonging to anti-cancer agents. Among them, there were antibiotics, non-steroidal anti-inflammatory drugs, anti-parasitic agents, hormones and their analogs, and antidepressants [1].

To begin with, phthalocyanine-sulfonamide (Pc-SA) conjugates bearing four or eight sulfonamide substituents were synthesized through direct fabrication of functionalized phthalonitrile derivatives, which were then subjected to cyclotetramerization to form the ZnPc macrocycles [49]. The sulfonamide moieties included both simple alkyl/aryl groups and more complex heterocyclic or long-chain alkyl derivatives (Figure 5).

These structural variations impacted their photodynamic antimicrobial efficiency. The conjugates were either used directly or encapsulated within polyvinylpyrrolidone (PVP) micelles to improve water solubility and delivery. Spectroscopic analysis revealed efficient singlet oxygen production with DPBF photodegradation yields between 59 and 91% in phosphate-buffered saline (PBS) and 92–100% in DMF for the PVP-encapsulated forms. The Q-band absorptions typical for phthalocyanines were maintained, although exact maxima were not specified in the study [49].

Indomethacin (IMC) is a nonsteroidal anti-inflammatory drug (NSAID) used to reduce fever, pain, stiffness, and inflammation, commonly in conditions like arthritis, gout, and bursitis. It works by inhibiting cyclooxygenase (COX) enzymes, which decreases the production of prostaglandins involved in inflammation and pain [50]. Huang et al. developed a novel zinc(II) phthalocyanine–indomethacin conjugate (IMC-Pc) to address issues of photosensitizer aggregation, tumor targeting, and intracellular localization [51]. The parent compound was a mono-phenoxy-4-carboxy-substituted zinc phthalocyanine. Synthesis involved functionalizing IMC with NHS (N-hydroxysuccinimide) to activate its carboxyl group, followed by amide bond formation with a hexamethylenediamine linker attached to the ZnPc (Figure 6, **XII**). This modular strategy ensured site-specific conjugation and maintained the photosensitizing core structure.

Spectroscopically, the conjugate displayed a Q-band at 674 nm in DMF, typical for ZnPc derivatives. The IMC unit, an electron-rich NSAID, did not adversely affect the photophysical properties of ZnPc and instead enhanced its biological targeting. Fluorescence spectroscopy demonstrated increased emission upon COX-2 binding, confirming a successful interaction and reduced aggregation. Biological assays showed that IMC-Pc generated significantly higher levels of ROS upon irradiation (λ = 670 nm) and demonstrated a potent, selective cytotoxic effect in HepG2 cancer cells while sparing normal HELF cells [51].

Another antibiotic used for conjugation was ciprofloxacin (CIP). In the study by Magadla et al., two amphiphilic asymmetric zinc(II) phthalocyanine–ciprofloxacin conjugates were synthesized to function as dual-action photodynamic antimicrobial chemotherapy (PACT) agents. The parent compounds were ZnPc substituted with benzo[d]thiazole-derived thiol or phosphonium groups, contributing to their amphiphilicity and facilitating bacterial uptake. Conjugation to CIP was achieved through N-acylation of the piperazine ring of CIP using a phenoxyethylcarboxy linker, activated with EDC and DMAP (Figure 6, **XIII**) [52].

Spectroscopically, the conjugates displayed strong Q-band absorptions at 684 and 687 nm, moderate fluorescence quantum yields in DMSO (Φ_F_ = 0.031 and 0.058), and high singlet oxygen quantum yields (Φ_Δ_ = 0.66 and 0.61), confirming their suitability for PDT. Bioelectrochemical techniques like square wave voltammetry and electrochemical impedance spectroscopy were employed to confirm uptake into biofilms of *E. coli* and *S. pneumoniae*, while Raman and TOF-SIMS analyses demonstrated intracellular ROS-mediated damage and effective penetration. Notably, both conjugates exhibited potent antibacterial effects under light and retained significant dark toxicity. Log reduction values reached up to 9.2 for *E. coli* and 7.23 for *S. pneumoniae*, highlighting the therapeutic potential of Pc-CIP conjugates in combating biofilm-associated infections [52].

Isoniazid (INH) is an antibiotic primarily used to prevent and treat tuberculosis (TB). It works by inhibiting the synthesis of mycolic acids, essential components of the mycobacterial cell wall, making it highly effective against *Mycobacterium tuberculosis* [53]. In the study by Nkanga et al. [54], a pH-responsive drug delivery system was developed by conjugating isoniazid, a hydrophilic anti-tubercular drug, to a hydrophobic ZnPc via a hydrazone bond to form a ZnPc–INH conjugate (Figure 6, **XIV**). The parent Pc compound used was a tetra-substituted derivative bearing peripheral phenoxybenzaldehyde groups. The hydrazone linkage was formed using acetic acid as a catalyst, allowing the pH-sensitive release of INH [54].

Optical analysis in DMSO solution showed a Q-band at 681 nm, indicating a slight red shift in absorbance after conjugation, confirming the structural modification. The conjugate was then encapsulated into liposomes using the film hydration method. The resulting ZnPc–INH-loaded liposomes exhibited favorable characteristics with an average particle size of 506 nm, a zeta potential of −55 mV, and an encapsulation efficiency of 72%. Dialysis studies demonstrated a pH-dependent INH release profile, with minimal release at neutral pH (22% at pH 7.4) and nearly complete release in acidic environments (100% at pH 4.4), making the system particularly suitable for macrophage-targeted delivery and site-specific drug release in TB therapy [54].

In the follow-up study by Nkanga et al. in 2019 [55], the researchers addressed the economic and scalability challenges of liposomal anti-tubercular drug delivery by employing a cost-effective, organic solvent-free heating method (HM) to encapsulate previously mentioned isoniazid-phthalocyanine conjugates complexed with γ-cyclodextrin (γ-CD) into liposomes. γ-CD inclusion was designed to enhance encapsulation efficiency without relying on organic solvents. Complexation was confirmed through UV–Vis, magnetic circular dichroism, ^1^H NMR, DOSY, and FT-IR spectroscopy. Notably, in DMSO solution, the Q-band intensity of ZnPc increased upon CD complexation, while the B-band at 320 nm decreased with increasing CD content. Fluorescence properties also improved, with Φ_F_ rising from 0.129 to 0.179 and τF from 2.865 to 2.891 ns. Despite a decrease in singlet oxygen yield from 0.837 (free ZnPc-INH) to 0.601 (CD-treated), encapsulation remained efficient (71%), with resulting liposomes displaying favorable characteristics: particle size of 240 nm, zeta potential of −57 mV, and long-term stability. Release studies revealed pH-sensitive behavior with up to 100% drug release at pH 4.4, confirming the potential of this CD-complexation-assisted liposomal platform for site-specific tuberculosis therapy [55].

The next drug used was mestranol. Mestranol is a synthetic estrogen used in some combined oral contraceptives. It is a prodrug that is converted in the liver to ethinylestradiol, its active form, which helps regulate hormone levels to prevent ovulation [56]. In the study by Nováková et al., an unsymmetrical ZnPc bearing three tert-butylsulfanyl groups on its periphery was functionalized with an azide moiety to serve as a modular platform for Cu(I)-catalyzed azide-alkyne cycloaddition (“click” chemistry). The coupling was performed via triazole linkage formation using a propyl arylamide linker (Figure 6, **XV**) [57].

Spectroscopic analysis revealed that the ZnPc–mestranol conjugate retained high singlet oxygen generation efficiency (Φ_Δ_ = 0.84 in pyridine, compared to 0.80 for the parent Pc) and exhibited a strongly red-shifted Q-band in pyridine at 727 nm, favorable for deep tissue penetration in photodynamic therapy. Importantly, the click-derived conjugation did not impair the excited-state behavior or photosensitizing properties of the phthalocyanine core. The addition of mestranol was expected to enhance tumor targeting, especially in hormone-responsive tissues, due to potential interactions with estrogen receptors [57].

In a study by Zhao et al. performed in 2022, a new class of phthalocyanine–artesunate conjugates was developed to address the current limitations in sonodynamic therapy (SDT), particularly the lack of potent and efficient sonosensitizers. Artesunate (ARS), a semi-synthetic derivative of artemisinin and a widely used antimalarial agent, was conjugated via esterification to zinc(II) phthalocyanines (ZnPcTs) modified with triethylene glycol chains, yielding amphiphilic conjugates ZnPcT_1_A, ZnPcT_2_A, and ZnPcT_4_A, possessing one, two, or four ARS moieties, respectively. The synthesis involved an EDCl/DMAP-catalyzed coupling of artesunate with mono-, di-, and tetra-substituted ZnPcTs (Figure 6, **XVI**) [58]. Among the conjugates, ZnPcT_4_A demonstrated the most significant sonodynamic activity, generating approximately 10 times more ROS under ultrasound irradiation than the reference sonosensitizer protoporphyrin IX (PPIX). Strikingly, the authors reported the first observation of aggregation-enhanced sonodynamic activity (AESA), whereby the ROS generation of ZnPcT_4_A in its aggregated state was up to 60 times higher than in its disaggregated form. This enhancement was attributed to increased ultrasonic cavitation of the nanostructures formed in aqueous environments. The AESA effect appeared closely linked to both the molecular aggregation tendencies of the conjugates and the resulting particle sizes of the self-assembled structures. ZnPcT_4_A also exhibited strong antitumor efficacy and good biocompatibility. In vitro cytotoxicity tests revealed that sonocytotoxicity under light irradiation λ ≥ 610 nm (IC_50_ = 2.0 μM) was substantially more potent than photocytotoxicity (IC_50_ = 7.6 μM). Furthermore, when sonodynamic and photodynamic treatments were combined (SPDT), the cytotoxic effect was further enhanced (IC_50_ = 0.7 μM), indicating a synergistic therapeutic response. Tumor inhibition studies confirmed the therapeutic potential, with ZnPcT_4_A achieving up to 98% inhibition of tumor growth [58].

Moclobemide (MLB) is a reversible inhibitor of monoamine oxidase A (RIMA) used to treat depression and social anxiety disorder. It works by increasing the levels of serotonin, norepinephrine, and dopamine in the brain, with fewer dietary restrictions and side effects compared to traditional MAO inhibitors [59]. Also, in 2022, a novel phthalocyanine-based photosensitizer, ZnPc-MLB, was rationally designed to target monoamine oxidase A (MAO-A), an enzyme known to be overexpressed in prostate cancer. The conjugate was synthesized through Cu(I)-catalyzed azide–alkyne cycloaddition (“click” chemistry), coupling a mono-alkyne tetraethylene glycol-substituted ZnPc with MLB (Figure 6, **XVII**) [60]. The resulting conjugate retained a strong Q-band at 677 nm in DMF and a singlet oxygen quantum yield (Φ_Δ_) of 0.61, closely matching that of the parent Pc (Φ_Δ_ = 0.62), which is higher than unsubstituted ZnPc (Φ_Δ_ = 0.56). ZnPc-MLB demonstrated markedly enhanced uptake in MAO-A-overexpressing prostate cancer cells (DU145 and PC-3), particularly notable at the 8-h incubation point, compared to the reference compound Pc-TEG. Mitochondrial localization studies and molecular docking further supported the high binding affinity of ZnPc-MLB for the MAO-A enzyme. Functionally, ZnPc-MLB exhibited potent photodynamic cytotoxicity in vitro, with IC_50_ values (λ = 670 nm) of 8.3 nM against DU145 cells and 84 nM against PC-3 cells, significantly lower than those observed for Pc-TEG. Additionally, ZnPc-MLB showed a promising ability to inhibit cancer cell migration and invasion, as demonstrated by wound healing and transwell assays, suggesting potential to limit metastasis [60].

## 5. Phthalocyanines Conjugated with Targeting Biomolecules

### 5.1. Folic Acid

Using folic acid (FA) for targeting tumor cells represents one of the most promising and actively developing approaches for targeted delivery of photosensitizers to tumors [61]. Unquestionable advantages of the use of FA include the following: high availability and low cost, biocompatibility, possibility of use in different targeted delivery systems (direct conjugation with photosensitizer via linker, conjugation with liposomes or nanoparticles), hyperexpression of the FA receptor on the surface of many tumor cells including brain, lung, ovaries, breast, colon, prostate, nose, and throat tumors and hematopoietic malignancies of myeloid origin such as chronic and acute myelogenous leukemia [62]. Furthermore, the undifferentiated highly metastatic tumors express much more FA receptors than their localized analogs with low-grade malignancy. Hyperexpression of the FA receptor has not been observed in the cells of normal tissues, with the exception of activated macrophages in some inflammatory diseases. All these allow using FA as a vector for targeted delivery of photosensitizer to the tumor [63]. FA has been frequently employed in Pc-based conjugates as it possesses terminal carboxylic and amino groups, enabling it to form amide bonds with phthalocyanines with peripheral amino or carboxylic groups, respectively. For instance, in the study by Lin et al., a novel zinc(II) phthalocyanine–folate conjugate, PcN-FA, was synthesized and designed to self-assemble into an activatable nanophotosensitizer (NanoPcN-FA) for targeted PDT. A control system lacking folate modification (NanoPcN) was used to compare targeting specificity and photoactivity. The PcN-FA conjugate was prepared by coupling a peripheral amine-substituted ZnPc to folic acid via a tyramine linker using amide bond formation (Figure 7, **XVIII**) [64].

While PcN-FA exhibited strong photoactivity in organic solvents (e.g., DMF), its self-assembled nanoparticulate form, NanoPcN-FA, demonstrated significantly quenched photophysical properties in aqueous environments, showing a dramatic 96% reduction in singlet oxygen generation, as confirmed by very slow DPBF degradation. This “super quenching” effect also included suppression of fluorescence [64]. On the other hand, in vitro assays revealed that NanoPcN-FA was selectively taken up by folate receptor (FR)-overexpressing SKOV3 ovarian cancer cells via a receptor-mediated endocytosis pathway. This targeting mechanism was validated through competitive uptake experiments, where the presence of free folic acid significantly reduced NanoPcN-FA internalization and photocytotoxicity. Notably, the IC_50_ under light irradiation (λ > 610 nm) of NanoPcN-FA was 0.75 μM in SKOV3 cells, which rose to 2.10 μM in the presence of excess folate, while NanoPcN showed no significant change (1.12 μM vs. 0.89 μM), further supporting FR-specific targeting. Importantly, the quenched photodynamic activity of NanoPcN-FA was restored intracellularly following uptake by FR-positive cells, leading to effective ROS generation and photocytotoxicity. In vivo studies using S180 tumor-bearing mice demonstrated that the folate-functionalized nanoparticles preferentially accumulated at the tumor site and achieved an impressive tumor growth inhibition rate of 95% after light activation [64].

Another synthetic approach to create ZnPc-FA conjugates was used by Matlou et al. In this study, two unsymmetrical ZnPc derivatives, mono carboxyphenoxy phthalocyanine and its tri-*t*-Bu derivative, were successfully conjugated to FA via amide bond formation. The conjugation was achieved using the carboxyl-functionalized phthalocyanines and the amino group of FA, employing DCC and DMAP as coupling agents (Figure 7, **XIX**) [65].

Structural confirmation of the resulting conjugates, designated 1-FA and 2-FA, was obtained through FTIR, mass spectrometry, NMR, and elemental analysis. Compared to physical mixtures of phthalocyanines and folic acid, the chemically linked conjugates demonstrated markedly enhanced water solubility, a key feature for biological and photodynamic applications. This increased solubility enabled accurate evaluation of their singlet oxygen generation capacity in aqueous media [65]. The Φ_Δ_ of the conjugates was notably high in DMSO - 0.61 for 1-FA and 0.47 for 2-FA and, while lower in water, they still retained measurable activity (0.17 and 0.12, respectively). These values surpassed those of the corresponding physical mixtures, underscoring the improved photophysical performance resulting from the covalent attachment of FA. Importantly, the conjugation did not significantly affect the absorption characteristics (Q-band position) of the parent phthalocyanines, indicating that the photodynamic functionality of the core ZnPc was preserved [65].

The same research team employed mono-cinnamic acid-substituted zinc(II) phthalocyanine to synthesize the ZnPc-FA conjugate through amide bond formation with DCC and DMAP (Figure 7, **XX**) [66]. The physicochemical, optical, and biological activities of this conjugate were compared with those revealed by a previous conjugate (Figure 7, **XIX**) and amine-functionalized magnetic nanoparticles (AMNPs).

The resulting FA-conjugated phthalocyanines were water-soluble, in contrast to the parent hydrophobic compounds, enabling their photophysical characterization and biological evaluation in aqueous environments. UV–Vis analysis revealed red-shifted Q-bands in water (∼690 nm) compared to DMSO (∼675 nm), consistent with the solvation environment. Singlet oxygen quantum yield measurements showed that while Φ_Δ_ decreased slightly for monocinnamic acid-terminated ZnPc-FA conjugate compared to the parent compound (from 0.48 to 0.41), it increased for the previously described conjugate (from 0.56 to 0.61), indicating that conjugation with FA did not compromise the photodynamic potential [66]. Biological evaluations using human MCF-7 breast cancer cells demonstrated that both FA- and AMNP-conjugated complexes exhibited reduced dark toxicity compared to the unmodified phthalocyanines. However, under light irradiation, ZnPc-FA conjugates showed superior PDT activity relative to their AMNP-conjugated counterparts, likely due to enhanced cellular uptake mediated by folate receptor targeting. At the highest tested concentration (80 μg/mL), the FA conjugates resulted in less than 40% cell viability, compared to higher survival rates observed for the AMNP formulations [66].

In another study, Ogbodu et al. investigated the photodynamic therapeutic efficacy of a folic acid directly conjugated to zinc monoamino phthalocyanine (ZnMAPc-FA), both in its free form and when immobilized onto single-walled carbon nanotubes (SWCNTs), resulting in the ZnMAPc-FA-SWCNT conjugate [67]. A control compound, SWCNT-FA (without the photosensitizer), was also included to assess the individual contribution of the nanocarrier. ZnMAPc, bearing a peripheral amino group, was covalently linked to folic acid via amide bond formation and subsequently immobilized onto SWCNTs. The resulting formulations were evaluated for cytotoxicity and photodynamic activity against the melanoma A375 cell line across a concentration range (5–80 µM) [67]. All compounds exhibited negligible dark toxicity at all tested concentrations, confirming their safety in the absence of light. However, upon irradiation with a 676 nm diode laser at a light dose of 5 J/cm^2^ and power density of 98 mW/cm^2^, both ZnMAPc-FA and ZnMAPc-FA-SWCNT showed substantial phototoxicity, inducing approximately 60% and 63% cell death, respectively, at 10 µM. In contrast, SWCNT-FA exhibited minimal photodynamic or photothermal effects, with only 23% cell death observed post-irradiation [67].

The same type of conjugation was used by Wang et al. [68], who report the development of a folic acid-conjugated zinc tetraaminophthalocyanine photosensitizer designed to enhance both PDT of cancer, with a focus on folate receptor (FR)-targeting and two-photon excitation (TPE) capabilities. ZnPc, bearing four peripheral amino groups, was covalently linked to folic acid through an amide bond using DCC/NHS chemistry. Despite some aggregation issues in aqueous solution, ZnPc–FA demonstrated improved monomeric behavior under physiological conditions due to interactions with proteins, facilitating its use in biological environments [68]. The targeted nature of ZnPc–FA was confirmed through cellular uptake studies. It showed significantly enhanced uptake by FR-overexpressing KB cells (human nasopharyngeal carcinoma) compared to non-conjugated ZnPc. This effect was abolished when cells were pretreated with excess free FA, confirming the specificity of FA-mediated binding. Confocal fluorescence imaging further validated this selectivity, as ZnPc–FA showed negligible affinity for FR-negative A549 cells (human lung carcinoma) [68]. Photophysically, ZnPc–FA exhibited a greater two-photon absorption cross-section and higher singlet oxygen quantum yield than the clinically used AlPcS (sulfonated aluminum phthalocyanine), positioning it as a superior photosensitizer. Under 780 nm femtosecond laser excitation, ZnPc–FA produced bright fluorescence signals in KB cells and generated singlet oxygen in a dose-dependent manner. Functionally, ZnPc–FA achieved stronger PDT effects than AlPcS under standard one-photon excitation and was especially potent under TPE conditions. Upon TPE, its phototoxicity toward KB cells was approximately 10 times higher than that of AlPcS, reducing cell viability to below 20% after just 9 min of irradiation [68].

### 5.2. RGD Peptide

The cyclic RGD peptide (where RGD stands for the amino acid sequence: R-Arginine, G-Glycine, D-Aspartic acid) is a well-established tumor-targeting ligand that specifically binds to αvβ_3_ integrin receptors, which are overexpressed in various cancer cells, including glioblastoma, prostate, and breast cancer cells [69,70]. By leveraging this selective interaction, photosensitizers such as phthalocyanines can be conjugated with cyclic RGD peptides to achieve enhanced tumor selectivity, cellular uptake, and PDT efficiency while minimizing effects on healthy cells. This targeted approach has been increasingly explored in recent years, as detailed in several key studies.

In the work by Ke, a 1,4-di-PEG-substituted ZnPc was conjugated to the cyclic peptide cRGDfK via a triazole linker (Figure 8, **XXI**) [71]. This conjugate in DMF exhibited a high singlet oxygen quantum yield (Φ_Δ_ = 0.80), though its fluorescence emission was relatively weak. Importantly, the conjugate showed significantly higher uptake in αvβ_3_-positive U87-MG glioblastoma cells than in αvβ_3_-negative MCF-7 breast cancer cells, confirming the targeting ability conferred by the RGD moiety. Interestingly, despite the enhanced uptake, the photocytotoxicity toward both cell lines was comparable, likely due to similar ROS generation efficiencies. Confocal microscopy revealed that the conjugate primarily localized in lysosomes, not in other organelles such as the nucleus or mitochondria [71].

Another contribution in this field was made by Luan, who synthesized an unsymmetrical ZnPc bearing both 4-tert-butylphenol and cRGDyK moieties (Figure 8, **XXII**) conjugated via amide bond formation between the aspartic acid (amine group) and butoxycarboxylic unit of Pc [72]. Despite its high singlet oxygen quantum yield in DMF (Φ_Δ_ = 0.55), the conjugate displayed negligible phototoxicity under light exposure λ ≈ 660 nm up to 12 J/cm^2^ and showed no dark cytotoxicity. Nonetheless, the compound achieved enhanced cellular uptake in αvβ_3_-positive DU145 and PC3 cancer cells and exhibited strong mitochondrial fluorescence [72]. In a follow-up study, the same conjugate demonstrated potent photocytotoxicity in vitro with an IC_50_ of just 0.04 µM (λ ≈ 660 nm) in DU145 prostate cancer cells, indicating a substantial therapeutic effect. This performance was attributed to its threefold higher cellular uptake compared to the non-targeted analog, confirming that the RGD conjugation not only enhances selectivity but can significantly improve PDT efficacy when internalization and ROS generation are efficiently combined [73]. Collectively, these studies illustrate that while RGD conjugation may not drastically alter the core photophysical properties of phthalocyanine, it significantly improves tumor selectivity and cellular internalization, especially in αvβ3 integrin-overexpressing cells.

Some tumor cells overexpress receptors of the epidermal growth factor (EGF). EGF receptor (EGFR) is a transmembrane protein with tyrosine kinase activity. Upon binding with the surface receptor, the EGF derivatives are engulfed by the cells via the receptor-mediated endocytosis, which provides their intracellular accumulation [63]. In terms of targeted-PDT, EGF peptide can be utilized as a binding moiety that enhances the bioavailability of the synthesized conjugate. Chu and colleagues developed a streamlined one-pot synthetic method for simultaneously cyclizing peptides and conjugating them to phthalocyanines. This approach utilized a specially designed bifunctional linker containing both a bis(bromomethyl)benzene unit and a cyclopentadiene moiety. The synthetic strategy involves two key steps: first, the site-selective nucleophilic substitution of the linker with two cysteine residues from a linear peptide, facilitating in situ cyclization; and second, the resulting intermediate undergoes a Diels–Alder reaction with a maleimide-functionalized ZnPc, forming a stable conjugate (Figure 9) [74].

This method proved efficient for producing three distinct cyclic peptide–phthalocyanine conjugates, each obtained in moderate isolated yields (20–26%). Importantly, one of these conjugates incorporated a cyclic epidermal growth factor receptor (EGFR)-binding peptide with the sequence CMYIEALDKYAC. This particular conjugate exhibited enhanced biological performance: it demonstrated selective uptake in EGFR-positive cancer cell lines (HT29 and HCT116), while minimal uptake was observed in EGFR-negative lines (HeLa and HEK293). As a result, the conjugate showed significantly higher photocytotoxicity in EGFR-positive cells. Further validation came from in vivo studies, where intravenous administration of the conjugate to HT29 tumor-bearing nude mice led to preferential tumor accumulation and effective tumor growth inhibition upon photodynamic treatment [74].

### 5.3. Biotin

Biotin, also known as vitamin B_7_, is a small, water-soluble molecule recognized for its strong affinity to biotin receptors, which are overexpressed in many cancer cells [75,76]. This makes biotin an attractive targeting ligand in PDT, enabling the selective delivery of photosensitizers to tumor tissues. Its low immunogenicity and ease of conjugation further enhance its suitability for targeted cancer therapies, potentially improving the efficacy and minimizing off-target effects of PDT agents [22].

Balçik-Erçin and co-workers developed a biotin-functionalized ZnPc derivative (Figure 10, **XXIII**) as a targeted photosensitizer for PDT against cancer cells. The compound was designed with a mono-biotin moiety (connected to the linker via amide bond formation) and three polyoxyethylene branches to enhance water solubility and promote selective cellular uptake [77]. For comparison, a non-biotinylated amino-functionalized analog was also synthesized. Both compounds exhibited comparable photophysical and photochemical properties in DMSO, including similar Q-band absorption features. The polyoxyethylene substitution resulted in an 11 nm red shift relative to unsubstituted ZnPc in DMSO, while the addition of the biotin moiety did not significantly alter the absorption wavelength. Singlet oxygen quantum yields for compounds were both higher than that of unsubstituted ZnPc, suggesting efficient ROS generation. The key distinction emerged from biological evaluations in HeLa (cervical carcinoma) and HuH-7 (hepatocellular carcinoma) cell lines. **XXIII** demonstrated substantially higher cellular uptake (34.5-fold in HuH-7 and 459-fold in HeLa cells) compared to the non-biotinylated derivative, attributed to the biotin-mediated targeting. Subcellular localization revealed cytoplasmic accumulation, and flow cytometry analysis (Annexin V/PI staining) showed dramatic increases in late apoptosis under 690 nm (± 10 nm) light irradiation (63.9% in HeLa and 83.6% in HuH-7), compared to negligible levels in the dark. In vitro colony formation assays confirmed the strong phototoxic effect of **XXIII** following light exposure. The biotin conjugation was therefore found to significantly enhance PDT efficacy, likely through increased cancer cell targeting, uptake, and singlet oxygen production [77].

In the second study related to biotin conjugation, Okoth et al. reported the synthesis and photodynamic evaluation of an isothiocyanate-functionalized ZnPc, which served as a versatile scaffold for bioconjugation via thiourea linkage. Starting from a p-aminophenoxy-substituted Pc, the isothiocyanato-Pc derivative was obtained and subsequently reacted with nucleophilic amines, ethanolamine, biotin hydrazine, and biotin ethylenediamine (**L**), to afford three novel conjugates in 60–75% yields under mild conditions (room temperature in DMF or DMSO with triethylamine) (Figure 10, **XXIV**) [78]. All conjugates exhibited typical phthalocyanine photophysical behavior in DMF, with strong Q-band absorption at ~677 nm, emission centered at ~683 nm, and fluorescence quantum yields ranging from 0.18 to 0.27 in DMF. Although the biotin–Pc conjugates demonstrated slightly lower fluorescence efficiency, possibly due to enhanced molecular flexibility increasing non-radiative decay, they maintained effective photodynamic activity. In vitro, the conjugates showed no dark toxicity up to 200 μM but induced significant phototoxicity upon light activation (IC_50_ ≈ 7 μM at 1.5 J/cm^2^) in HEp2 human carcinoma cells. Confocal microscopy revealed subcellular localization in the lysosomes, Golgi apparatus, and endoplasmic reticulum, indicating broad intracellular distribution. One biotin-functionalized conjugate (n = 0) was further evaluated in vivo in nude mice bearing HT-29 tumors. Following intravenous administration, the conjugate selectively accumulated in tumor tissues within 6 h, highlighting its tumor-targeting capability [78].

### 5.4. Carbohydrates

The activity of many metabolic pathways, including the ones associated with carbohydrate transformations, changes in tumor cells, is likely due to accelerated metabolism and increased energy demand by the actively metabolizing tumor cells. The Warburg effect and high expression of GLUT1 in tumor cells determine a high rate of glucose uptake, which could be used for targeted delivery. It was shown in a number of studies that the accumulation of photosensitizers in tumor cells increases after conjugation with carbohydrates [63].

Pursuing the goal of obtaining phthalocyanines conjugated with carbohydrates, two series of amphiphilic carbohydrate–nickel(II) Pc conjugates were synthesized using either direct glycosylation or Cu(I)-catalyzed azide–alkyne “click” reactions (Figure 11, **XXV**) [79]. The starting material, a hydroxylated phthalocyanine, was glycosylated with perbenzoylated trichloroacetimidate donors or converted to an azido derivative, followed by coupling with propargylated sugars (glucose, galactose, mannose, and lactose). A tetraethylene glycol spacer and triazole linkers were introduced to facilitate conjugation and maintain hydrophilic-lipophilic balance. The click chemistry pathway offered high regioselectivity and excellent yields (>90%), except in the lactose series (~70%) due to purification difficulties. The final compounds featured six lipophilic hexylthio chains and hydrophilic sugar units, imparting self-assembling properties. UV–Vis analysis confirmed characteristic Q-band absorption in chloroform at ~695 nm, largely unaffected by the type of sugar or the triazole ring, indicating minimal disruption of the phthalocyanine’s photophysical core. However, lactose-conjugated phthalocyanines exhibited broadened Q-bands due to aggregation via intermolecular hydrogen bonding, suggesting dimer formation. While the sugar identity had little effect on absorption properties, purification, particularly for lactose derivatives, proved challenging due to the formation of by-products. Additionally, strong aggregation and low solubility limited NMR and photophysical analysis [79].

In 2012, glycodendritic conjugates of perfluorinated porphyrin (TPPF_20_) and phthalocyanine (ZnPcF_16_) were synthesized by Silva et al. to enhance water solubility and photodynamic efficacy [80]. The approach involved creating galactose-rich dendrons via a two-step synthesis: first, a selective di-substitution of 2,4,6-trichloro-1,3,5-triazine (TCT) with galactose moieties, followed by linking with 1,3-dimercaptopropane to form flexible dendritic scaffolds. These were subsequently conjugated to the porphyrinoid cores via nucleophilic aromatic substitution, yielding highly galactosylated products: 8 sugars per porphyrin and 16 per phthalocyanine molecule. Acid hydrolysis removed protective isopropylidene groups, yielding hydrophilic, biologically relevant conjugates (Figure 11, **XXVI**). The resulting conjugates, both porphyrin- and phthalocyanine-based, displayed excellent solubility in polar organic solvents (DMF, DMSO) and reasonable solubility in aqueous buffer. UV–Vis spectroscopy confirmed monomeric behavior in DMF, with well-defined Soret (∼415 nm) and Q-bands. Minimal aggregation was observed in phosphate-buffered saline, with Beer–Lambert linearity up to 19 µM (P) and 9 µM (Pc), affirming their suitability for biomedical applications. Both compounds exhibited moderate fluorescence (Φ_F_ = 0.09–0.13 in DMF) and high singlet oxygen quantum yields, comparable to benchmark photosensitizers. Notably, both conjugates bound effectively to human serum albumin (HSA), supporting their potential in systemic drug delivery. The only notable challenge was precipitation of the phthalocyanine derivative in DMSO, forming crystalline needles, which was an issue that may affect formulation strategies [80].

One year later, Lafont and co-workers investigated the photodynamic activity of monoglycoconjugated ZnPcs, focusing on how carbohydrate type (galactose, mannose, lactose) and conjugation method (glycosylation vs. click chemistry) influence biological performance. Two sets of water-soluble ZnPc derivatives were synthesized: one via classical glycosylation and the other using Cu(I)-catalyzed azide–alkyne “click” reactions, generating a 1,2,3-triazole linkage between the phthalocyanine and sugar. Both series employed a tetraethylene glycol spacer to enhance hydrophilicity (Figure 11, **XXVII** and **XXVIII**) [81]. UV–Vis data confirmed that triazole formation did not alter the photophysical properties of the ZnPc core significantly. However, solubility issues plagued NMR characterization of deprotected compounds, especially for 1-Sug derivatives. Photodynamic efficiency was assessed in HT-29 human colon adenocarcinoma cells, with LD_50_ (median lethal dose) values measured to quantify cell killing potency. The mannose-conjugated ZnPc via glycosylation demonstrated the highest activity under red light λ > 600 nm irradiation (LD_50_ = 110 μM), despite showing the lowest intracellular uptake, revealing a surprising disconnect between uptake and therapeutic effect. In contrast, triazole-linked conjugates (click-derived) showed reduced biological activity, suggesting that the added rigidity and steric bulk of the triazole may hinder recognition by cellular receptors. Notably, lactose derivatives exhibited poor PDT performance, possibly due to limited receptor binding on HT-29 cells. The study highlights how linker chemistry and sugar identity critically influence PDT efficacy, demonstrating that triazole linkages may compromise activity despite conferring synthetic advantages [81].

In a study by Yilmaz et al., saccharide-functionalized copper phthalocyanines (CuPc) and their metal-free precursors (H_2_Pc) were synthesized via Cu(I)-catalyzed azide–alkyne “click” reactions, aiming to evaluate their photoluminescence properties. The metal-free H_2_Pc was conjugated with 2-acetamido-2-deoxy-β-D-glucopyranosyl azide, forming a triazole linkage on each benzo group. During the reaction in THF/H_2_O, in situ complexation with copper occurred, yielding the saccharide–CuPc derivative with a 76% yield (Figure 11, **XXIX**) [82]. Spectroscopic analysis confirmed successful conjugation and metallation. Photophysical studies showed dramatic differences between the H_2_Pc and its Cu-complexed counterpart. The H_2_Pc derivative displayed strong red emission (λ_em_ = 701 nm in DMF), a high photoluminescence quantum yield (Φ_F_ = 0.42), and a long excited-state lifetime (3.89 ns). However, after copper coordination and saccharide derivatization, the emission blue-shifted to 684 nm, the quantum yield dropped to 0.14, and the lifetime decreased to 1.34 ns in DMF solution. The decline in luminescent efficiency was attributed to metal–ligand energy transfer and the introduction of weak electron-donating saccharide groups, which altered the electronic structure of the macrocycle. Additionally, a mixture of constitutional isomers was formed during synthesis, which could not be separated by HPLC, complicating purity and analysis. Despite reduced quantum efficiency, the conjugates remained soluble in polar solvents (THF, DMF, DMSO) and retained visible light emission, indicating potential for optoelectronic applications [82].

In another study concerning carbohydrates covalently conjugated to ZnPc, Mori et al. developed a fluorinated ZnPc–galactose conjugate as a novel photosensitizer for PDT, aiming to address limitations of conventional agents like Photofrin^®^, which suffer from weak absorbance beyond 630 nm. The design integrated 4 galactopyranosyl substituents to enhance water solubility and 12 peripheral fluorine atoms to fine-tune amphiphilicity, improving both bioavailability and tumor targeting. The synthetic approach involved tetramerization of galactopyranosylated trifluorophthalonitriles, followed by acidic deprotection. No additional linker was used (Figure 11, **XXX**) [83]. The resulting compound exhibited strong absorption in the 600–750 nm range in DMSO, with fluorescence maxima at 676 and 701 nm, suitable for deep-tissue light activation. Though the fluorescence quantum yield decreased from 0.37 in the non-fluorinated analog to 0.09 in **XLIV**, the conjugate remained effective due to high singlet oxygen generation and light-triggered cytotoxicity. Importantly, **XLIV** demonstrated strong photodynamic activity, killing ~90% of HT-1080 cancer cells upon 664 nm light irradiation, compared to only 35% for the non-fluorinated counterpart. This marked improvement was attributed to the fluorine-induced modulation of lipophilicity, aiding cellular uptake and membrane interaction. However, challenges included aggregation in aqueous media and dark cytotoxicity of acetal-protected intermediates, which rendered some precursors unusable [83].

Last but not least, in 2024, Makuch et al. investigated sugar-conjugated ZnPcs as targeted photosensitizers for PDT in psoriasis treatment. Both protected (acetylated) and unprotected glucose and galactose derivatives were synthesized via glycosylation and macrocyclization without spacers [84]. Among these, protected glucose–ZnPc (Glu-4-ZnPc-P) exhibited the most potent photocytotoxicity against IL-17A-stimulated HaCaT keratinocytes (IC_50_ = 2.55 µM, red light dose of 0.9 J/cm^2^), outperforming unprotected glucose and galactose conjugates, as well as free ZnPc. This activity was linked to enhanced mitochondrial localization and higher intracellular ROS generation upon illumination. Interestingly, although unprotected Glu-4-ZnPc showed greater cellular uptake (56.3% increase under IL-17A stimulation), it was significantly less phototoxic, suggesting that intracellular localization and molecular configuration play a more decisive role in PDT efficacy than uptake alone. UV–Vis analysis revealed sharp Q bands at 687 nm in DMSO, while in water, aggregation led to blue-shifted and split Q bands. In vivo application of Glu-4-ZnPc-P significantly reduced Psoriasis Area and Severity Index (PASI) scores, inhibited splenomegaly, and restored healthy skin morphology in an IMQ-induced mouse model. These results support the selective accumulation of sugar–ZnPc conjugates in hyperproliferative keratinocytes, likely via glucose transporters such as GLUT1. Despite challenges such as aggregation and limited aqueous solubility, this study highlights Glu-4-ZnPc-P as a promising antipsoriatic PDT candidate, providing a targeted, cytokine-responsive approach to psoriasis treatment with superior efficacy and manageable limitations [84].

### 5.5. Amino Acids

Amino acids can function as effective targeting units in photosensitizers due to their inherent biocompatibility and capacity to interact with specific cellular receptors or transporters [85]. Their conjugation to photosensitizers can enhance selective uptake by cancer cells or tissues, thereby increasing the precision of PDT [86]. Certain amino acids also promote cellular internalization or subcellular localization, further improving therapeutic outcomes. Beyond individual amino acids, their polymeric forms offer greater versatility by enhancing solubility, stability, and bioavailability [87]. These macromolecular structures enable multivalent interactions with cell surface receptors, facilitating targeted delivery and prolonged tumor retention. Additionally, polymeric amino acids can be engineered to respond to specific biological stimuli, allowing for controlled release and site-specific activation of photosensitizers within the tumor microenvironment.

Chen et al. developed a novel ZnPc–polylysine conjugate (ZnPc-PL) as a targeted antimicrobial photosensitizer for PACT in the treatment of periodontal disease (Figure 12, **XXXI**) [88]. This conjugate was designed to selectively accumulate in bacterial cells, particularly *Porphyromonas gingivalis*, the primary pathogen in periodontitis. By exploiting the cationic nature of poly-L-lysine, ZnPc-PL achieved superior bacterial uptake compared to anionic or neutral analogs, as well as to the previously studied Ce6-PL conjugate. Spectroscopic characterization revealed that ZnPc-PL possessed a Q-band at 680 nm and a Soret band at 344 nm in DMSO, enabling deep tissue penetration suitable for clinical applications. The ZnPc-to-polylysine ratio was approximately 1:1, as determined by the TNBS assay. In vitro phototoxicity assays using mammalian cells (BMSC and HPDLC) confirmed minimal toxicity, demonstrating high selectivity for bacteria over host cells. ZnPc-PL exhibited significantly improved photoinactivation of *P. gingivalis* in both in vitro and in vivo settings. In a beagle dog model of experimental periodontitis, PACT treatment with ZnPc-PL resulted in a 100-fold reduction in bacterial burden relative to controls (laser alone or untreated). Additionally, reductions in gingival crevicular fluid (GCF) volume and aspartate aminotransferase (AST) activity were observed, indicating decreased inflammation and cell damage [88].

Li et al. synthesized a series of ZnPc–oligolysine conjugates, ZnPc-(Lys)ₙ (n = 1, 3, 5, 7, 9), aiming to enhance water solubility and evaluate their potential as photosensitizers for PDT (Figure 12, **XXXII**) [89]. The oligolysine chains served both as solubilizing agents and as modulators of cellular uptake and photodynamic performance. All conjugates showed improved Φ_Δ_ relative to unsubstituted ZnPc, with values ranging from 0.60 to 0.74 (DMSO). Notably, ZnPc-(Lys)_7_ and ZnPc-(Lys)_9_ demonstrated the highest Φ_Δ_ values of 0.74. The conjugates were tested in both cancerous (BGC-823 human stomach adenocarcinoma) and normal (HELF human embryonic lung fibroblasts) cell lines. Cellular uptake showed a non-linear trend: ZnPc-(Lys)_5_ achieved the highest uptake in cancer cells, yet ZnPc-(Lys)_7_ exhibited the strongest photodynamic efficacy. Subcellular localization studies indicated preferential accumulation of ZnPc-(Lys)_3_ and ZnPc-(Lys)_9_ in mitochondria and lysosomes. A key methodological advancement in this study was the development of a real-time, impedance-based phototoxicity assay using electric cell-substrate impedance sensing (ECIS). This enabled continuous monitoring of cell viability following PDT treatment. The results revealed that ZnPc-(Lys)_7_ had the most pronounced photodynamic response, suggesting an optimal balance between photophysical properties and cellular interactions [89].

In another study, Nombona et al. developed and evaluated novel ZnPc–poly-L-lysine (ZnPc–PLL) conjugates, further hybridized with plasmonic nanoparticles (gold and silver), for PACT targeting *Staphylococcus aureus*. Two phthalocyanine conjugates were synthesized: ZnPc(SO_2_-PLL)_2_-ε-PL (2-ε-PL) and 4-tetrakis-(5-trifluoromethyl-2-pyridyloxy)phthalocyaninato Zn(II)-PLL (3-ε-PL) [90]. These conjugates were combined with gold (AuNPs) and silver nanoparticles (AgNPs) via amino group adsorption to form ZnPc–PLL–NP complexes. The 3-ε-PL conjugate, especially in combination with AgNPs, demonstrated the highest photodynamic activity. Upon 10 min of irradiation (fluence: 39.6 mW/cm^2^), bacterial growth was reduced to ~6% at a 3 μM drug dose. The minimum inhibitory concentration (MIC_50_) for 3-ε-PL in the presence of AgNPs was particularly low (< 0.0058 μM), highlighting the strong antibacterial efficacy even at low concentrations. Interestingly, despite producing less singlet oxygen than 2-ε-PL, 3-ε-PL achieved greater bacterial inactivation. This effect was attributed to its increased amphiphilicity and improved bacterial membrane interaction. The addition of silver nanoparticles significantly enhanced bacterial inhibition, indicating a synergistic relationship between the ZnPc–PLL conjugates and plasmonic nanoparticles [90].

Finally, Wang et al. in 2017 [91] synthesized and evaluated two arginine-functionalized ZnPc derivatives—ArgZnPc and ArgEZnPc—as photosensitizers designed to improve water solubility, cellular uptake, and PDT efficacy (Figure 12, **XXXIII** and **XXXIV**). Unlike conventional cationic photosensitizers that rely on quaternization or protonation, both of which have limitations such as instability at alkaline pH or requiring hazardous reagents, arginine’s guanidinium group maintains a stable positive charge over a broad pH range (5–9), offering a more biocompatible alternative. The conjugates were synthesized via EDC/HOBt-mediated amidation from a carboxylated ZnPc precursor [91].

Spectroscopic analysis showed characteristic Q-band absorptions in the 630–640 nm range (distilled water). However, fluorescence was diminished due to aggregation in aqueous solutions. Importantly, ArgEZnPc exhibited reduced aggregation and significantly higher singlet oxygen generation (ΦΔ = 0.402) compared to ArgZnPc (Φ_Δ_ = 0.07). Cellular experiments in HeLa cells revealed that ArgEZnPc achieved up to 96% uptake after 24 h and induced strong intracellular ROS production and phototoxic effects. Subcellular localization using organelle-specific trackers confirmed that both conjugates preferentially accumulated in lysosomes, with ArgEZnPc showing more pronounced lysosomal retention. In contrast, ArgZnPc demonstrated lower uptake and weaker photodynamic activity, likely due to internal salt formation between its guanidinium and carboxyl groups, which hindered interaction with cell membranes. Ultimately, ArgEZnPc was found to be a significantly more effective photosensitizer, reducing cell viability to approximately 13.8% upon 665 nm light activation, highlighting its potential as a potent and pH-stable PDT agent [91].

## 6. Phthalocyanines with Other Photosensitizers and Imaging Agents

### 6.1. BODIPY

BODIPY (boron-dipyrromethene) derivatives are a class of fluorescent dyes known for their high photostability, tunable photophysical properties, and strong absorption in the visible to near-infrared region. In PDT, BODIPY-based photosensitizers are increasingly explored due to their efficient singlet oxygen generation and the ability to target specific tissues or cellular compartments. These features make BODIPY derivatives promising candidates for improving the selectivity and efficacy of PDT in cancer treatment [92]. To increase the photodynamic activity of photosensitizers, these compounds were also employed in Pc-based conjugates, as shown by several researchers. In the 2014 study by Göl et al., a new series of asymmetrical ZnPc–BODIPY conjugates was synthesized to explore the already mentioned effect of increasing BODIPY substituents on photophysical properties relevant to PDT. The synthetic strategy involved a Pd-catalyzed Sonogashira coupling reaction between 2(3),9(10),16(17),23(24)-tetrakis(iodo) zinc(II) phthalocyanine (Iodo-Pc) and 4,4′-difluoro-8-(4-ethynyl)-phenyl BODIPY (Ethynyl-BODIPY), yielding ZnPc–BODIPY conjugates bearing one to four BODIPY units (Figure 13, **XXXV**) [93].

The introduction of BODIPY groups significantly modified the spectroscopic properties: Q-band absorptions shifted bathochromically (up to 709 nm in DMSO) and emission maxima progressively increased with each additional BODIPY unit (from 695 nm to 709 nm in DMSO). Although fluorescence quantum yields (Φ_F_) of the ZnPc core decreased with increasing BODIPY content compared to unsubstituted ZnPc (Φ_F_ = 0.20 in DMSO), singlet oxygen quantum yields (Φ_Δ_) improved substantially, reaching a maximum of 0.98 for the Pc–BODIPY conjugate, significantly higher than both Iodo-Pc (Φ_Δ_ = 0.54) and unsubstituted ZnPc (Φ_Δ_ = 0.67). Notably, the substitution of BODIPY units not only enhanced photophysical characteristics but also improved the solubility of otherwise poorly soluble Iodo-Pc derivatives. An efficient energy transfer from the excited BODIPY to the ZnPc core was confirmed, contributing to improved photosensitization. Despite minor reductions in fluorescence output under uniform excitation conditions, the conjugates showed promising photostability and light-harvesting efficiency, making them attractive for PDT applications. No biological data were reported, but the study provides a valuable foundation for future development of multifunctional PDT agents leveraging modular BODIPY–phthalocyanine architectures [93].

In another study on Pc-BODIPY conjugates, Osati et al. synthesized a range of BODIPY–phthalocyanine conjugates to investigate the effect of a substitution pattern on photophysical behavior and assess their potential in theranostic applications. ZnPc bearing iodo groups in either peripheral or non-peripheral positions was coupled to 1,3,5,7-tetramethyl-8-phenyl-BODIPY via Pd-catalyzed Sonogashira reactions, resulting in conjugates with linkages at β positions of the BODIPY ring (Figure 14, **XXXVI**) [94].

The resulting conjugates exhibited broad panchromatic absorption from the UV to the visible region due to the combined chromophoric systems. Spectroscopic data revealed efficient intramolecular energy transfer from the BODIPY units to the Pc core, confirmed by significant reduction in BODIPY emission upon conjugation and characteristic Pc fluorescence in the 700–720 nm range in THF. Some conjugates displayed split Q-bands and red-shifted absorptions depending on the relative positioning and dihedral angles between BODIPY and Pc moieties. This structural sensitivity provides a tunable platform for optimizing photophysical properties [94].

A copper-free Sonogashira cross-coupling reaction was utilized in the synthesis of a novel series of ZnPc–BODIPY conjugates as theranostic agents combining diagnostic imaging and PDT in a single molecule. Three types of conjugates were prepared: one with non-styryl BODIPY and two incorporating mono- and di-styryl BODIPY derivatives with electron-donating p-(N,N-diethylamino)benzaldehyde groups (Figure 15A, **XXXVII**) [95].

These conjugates featured extended absorption (500–719 nm in DMF) and emission properties (Φ_em_ ≈ 700 nm in DMF), attributed to efficient singlet–singlet energy transfer from the excited BODIPY donor to the phthalocyanine core, as confirmed by Förster resonance energy transfer behavior. Photophysical characterization showed enhanced fluorescence in acidic environments, making these conjugates promising for tumor imaging. However, conjugates with styryl substituents exhibited lower singlet oxygen generation due to intramolecular quenching and photoinduced electron transfer (PET) from the diethylamino group. ROS generation for the non-styryl conjugate remained comparable to parent ZnPc (Φ_Δ_ ≈ 0.56 in DMF), while photodegradation quantum yields varied from 1.10 × 10^−3^ to 3.10 × 10^−3^. Limited solubility and reduced emission in some derivatives highlighted challenges in balancing structural design with biological applicability [95].

In the follow-up studies, the authors developed two novel ZnPc–BODIPY conjugates featuring eight distyryl-BODIPY units per phthalocyanine core (Figure 15B, **XXXVIII** and **XXXIX**) [96]. These conjugates were synthesized via two complementary strategies: copper-free Sonogashira coupling (yielding ethynyl linkages) and Cu(I)-catalyzed “click” chemistry (producing triazole linkages). The BODIPY moieties were functionalized with electron-rich diethylaminophenyl groups to enhance photophysical responsiveness. The resulting conjugates displayed strong absorption across the visible spectrum and into the near-infrared (Q-bands at ~673–679 nm in DMF). Interestingly, while the compounds absorbed efficiently, they exhibited quenched fluorescence in DMF due to the already-mentioned PET phenomenon from the diethylamino groups. However, under acidic conditions, protonation of the amino groups suppressed PET, restoring fluorescence and demonstrating pH-dependent “turn-on” behavior suitable for tumor-selective imaging. Moreover, both conjugates generated singlet oxygen effectively under acidic pH, supporting their potential as pH-activated theranostic agents. Despite promising optical features, the conjugates suffered from poor solubility in common organic solvents (only soluble in DMF) and moderate photostability [96].

The last described study concerning ZnPc-BODIPY conjugates was performed by Ha et al. in 2020 [97], who synthesized a multifunctional phthalocyanine conjugate as a molecular “keypad lock”, capable of processing four different stimuli: GSH, acid, and two distinct wavelengths of light (>610 nm and 345 nm). The designed molecule incorporated three functional modules: a BODIPY dye (via acid-sensitive ketal linkage), a pyrene fluorophore (via a singlet oxygen-cleavable thioketal linker), and 2,4-dinitrobenzenesulfonate (DNBS) groups attached to the phthalocyanine core (Figure 16, **XL**) [97]. The synthesis involved multistep coupling strategies, including Knoevenagel condensation, click chemistry, and copper-catalyzed azide-alkyne cycloaddition to enable precise spatial control over energy transfer mechanisms.

In its intact form, the conjugate was non-fluorescent and photochemically inactive, due to active photoinduced electron transfer and blocked fluorescence resonance energy transfer (FRET). Activation required the correct sequence-specific input: GSH first removes DNBS groups, disrupting PET; then 345 nm light induces singlet oxygen production, cleaving the thioketal linker and enabling FRET between components, leading to strong fluorescence at 375 nm in PBS with Tween 80 and 0.2% DMF. Conversely, adding acid prematurely blocked this cascade, effectively keeping the system “OFF.” This sequence-dependency modeled a four-input molecular logic gate or “keypad lock”, with both fluorescence and singlet oxygen generation acting as output signals. UV–Vis analysis in DMF solution showed a B-band at ~340 nm and a Q-band at ~695 nm. Despite innovative logic behavior, the conjugate was only responsive under highly controlled input sequences, and its photoactivity was initially quenched, posing limitations for practical biological applications. Nonetheless, this work demonstrated a proof-of-concept molecular security system for programmable diagnostics and molecular computing platforms [97].

### 6.2. Porphyrinoids

Despite BODPIY derivatives, phorphyrinoid-based macrocyclic compounds such as porphyrins, chlorins, or their precursors (5-aminolevulinic acid, ALA) were also considered for obtaining Pc-based conjugates, leveraging the photoinduced electron transfer, enhancing PDT efficiency. The first examples were made by Soares et al. followed by studies of Hausmann et al. [98,99]. In a study focused on covalently NH-linked porphyrin–phthalocyanine (P–Pc) conjugates, two structurally distinct dyads were synthesized, differing by the attachment site of the porphyrin moiety - either through the meso-phenyl substituent or the β-pyrrolic position (Figure 17, **XLI** and **XLII**). The synthetic strategy combined the Buchwald–Hartwig cross-coupling of a phthalonitrile derivative with appropriately functionalized porphyrins, followed by cyclotetramerization to form the phthalocyanine core [98].

These structural variations resulted in notable changes in optical properties. Both conjugates exhibited a red-shifted Q-band absorption at approximately 704 nm in THF, about 23 nm beyond the parent phthalocyanine, indicating strong electronic interactions between the porphyrin and phthalocyanine units [98]. UV–Vis spectroscopy confirmed extended π-conjugation, while photophysical studies showed efficient intramolecular energy transfer from the porphyrin donor to the phthalocyanine acceptor, irrespective of the linkage position [98,99]. Fluorescence upconversion and transient absorption measurements revealed that the dyad linked via the β-pyrrolic position demonstrated superior S_1_ deactivation efficiency (∼90%) compared to the second one. Importantly, both conjugates were capable of singlet oxygen generation, an essential feature for photodynamic applications. Preliminary biological evaluations indicated that the conjugated systems exhibited improved performance compared to unmodified phthalocyanines, attributed to enhanced light absorption and more efficient photoinduced processes.

Methylpheophorbide a is a chlorophyll-a derivative commonly used as a precursor for synthesizing photosensitizers in PDT. It retains the porphyrin-like structure essential for light absorption and reactive oxygen species generation, making it valuable for biomedical and photochemical applications. A covalently linked conjugate of phthalocyanine and methylpheophorbide a was synthesized for the first time via a transesterification reaction between the α-keto methyl ester of methylpheophorbide a and free-base 2-(2-hydroxymethylbenzyloxy)-9(10),16(17),23(24)-tri-tert-butylphthalocyanine or its zinc(II) complex (Figure 17, **XLIII**) [100,101]. This transformation was carried out under mild conditions using iodine and DMAP as catalysts in toluene.

The resulting dyad exhibited panchromatic absorption in the UV–Vis spectrum (THF), with Q-bands observed at 667 nm (attributed to the pheophorbide unit) and 702 nm (shifted due to π-π stacking between the chromophores). Spectroscopic analysis using 1D and 2D NMR confirmed the presence of intramolecular π-stacking interactions, indicative of a compact and organized molecular architecture. Fluorescence studies showed energy transfer from the excited pheophorbide moiety to the phthalocyanine core, highlighting the dyad’s potential for photophysical applications. However, purification via chromatographic methods on SiO_2_ or Al_2_O_3_ significantly reduced the yield, likely due to enhanced oxidative degradation during the process [100].

In a study focused on advanced supramolecular hybrid systems, a novel series of fused porphyrin–phthalocyanine (P–Pc) conjugates was synthesized and investigated for their photophysical and structural properties [102]. The synthesis began with a Diels–Alder reaction between β-vinylporphyrin and fumaronitrile, yielding isomeric adducts that were subsequently dehydrogenated to form benzo[b]porphyrin-21,22-dicarbonitrile. This key intermediate underwent statistical cyclotetramerization with 4-tert-butylphthalonitrile in the presence of zinc acetate to generate the ZnP–ZnPc conjugate, while selective demetallation of the porphyrin moiety afforded H_2_P–ZnPc. The resulting dyads exhibit bent configurations due to steric hindrance yet maintain extensive π-conjugation and strong electronic coupling between the macrocycles (Figure 17, **XLIV**).

UV–Vis spectroscopy revealed significantly red-shifted absorption features and panchromatic behavior, making these conjugates effective light harvesters across the solar spectrum. Upon coordination with N-(4-pyridyl)fullero[c]pyrrolidine, supramolecular P–Pc–C_60_ triads were formed, which retained high absorption cross-sections. Photophysical studies demonstrated a sequential excitation deactivation pathway: initial energy transfer from porphyrin to phthalocyanine, followed by intramolecular charge transfer yielding photoinduced charge-separated states such as ZnP–(ZnPc)^•+^–(C_60_)^•−^. These systems also displayed potential for singlet oxygen generation due to their efficient light absorption and electron-transfer dynamics [102].

5-Aminolevulinic acid (5-ALA) is a precursor for the in situ synthesis of protoporphyrin IX (PPIX) inside living cells. In brief, two molecules of 5-ALA undergo condensation by the enzyme ALA dehydratase (also known as porphobilinogen synthase) to form porphobilinogen (PBG). Four PBG molecules are then linked together by hydroxymethylbilane synthase to generate hydroxymethylbilane, which spontaneously cyclizes into uroporphyrinogen III under the action of uroporphyrinogen III synthase. This intermediate is sequentially modified by uroporphyrinogen decarboxylase and coproporphyrinogen oxidase, leading to the production of protoporphyrinogen IX. Finally, protoporphyrinogen IX is oxidized by protoporphyrinogen oxidase to yield protoporphyrin IX.

A study by de Oliveira and co-workers reported the synthesis and characterization of two novel water-soluble phthalocyanine derivatives bearing 5-ALA covalently linked to their macrocyclic structures, aiming at potential PDT applications (Figure 17, **XLV**) [103]. The synthetic route involved a three-step process beginning with the preparation of a tetra-hydroxyethyloxy-substituted ZnPc from phthalonitrile using Zn(OAc)_2_ × 2H_2_O and N,N-dimethylethanolamine (DMAE), resulting in an isomeric mixture with high yield (78%). Following deprotection, the esterification of the free hydroxyl groups with ALA was carried out under mild acidic conditions, yielding the final conjugate in 44% over two steps. The acidic environment was crucial, as ALA is an ammonium salt susceptible to decomposition under basic conditions. Attempts to improve the esterification through activation with an acyl chloride derivative of ALA were unsuccessful due to concurrent degradation of the phthalocyanine core.

The final ALA-conjugated phthalocyanine displayed a strong Q band absorption at 682 nm in DMSO and a singlet oxygen quantum yield of 0.52, indicating efficient triplet-state population and promising photosensitizing potential [103].

In a follow-up study by Pavani et al., a phthalocyanine–5-aminolevulinic acid conjugate was developed to exploit a dual-mode PDT mechanism, combining the intrinsic photosensitizing properties of the phthalocyanine core with the intracellular biosynthesis of protoporphyrin IX from the released 5-ALA [104]. The conjugation was achieved via esterification between tetra-hydroxyethyloxy-substituted ZnPc and ALA, yielding a conjugate with a Q band centered at 682 nm in DMSO and a high singlet oxygen quantum yield of 0.69, indicating strong potential for ROS generation. Upon cellular uptake, the compound undergoes hydrolysis, releasing 5-ALA, which is metabolized to PPIX within the cell, enabling a synergistic phototoxic effect through simultaneous excitation of both phthalocyanine and PPIX using red and blue light, respectively. This dual sensitization significantly enhanced photoinduced cytotoxicity compared to the parent compound lacking ALA. Notably, the conjugate showed time-dependent increases in efficacy (blue light 408 ± 10 nm with red LED 639 ± 10 nm concomitantly): after 24 h of incubation, it achieved an IC_50_ value of 22 μM—more than twice as effective as the 3-h incubation (IC_50_ = 48 μM)—which correlated with improved cellular uptake (93% vs. 57%). These findings support the strategy of designing phthalocyanine-5-ALA hybrid photosensitizers as a promising route to improve PDT outcomes through combined photophysical and biochemical mechanisms [104].

### 6.3. Other Fluorescent Dyes

In terms of other dyes covalently conjugated to ZnPc, Kuznetsova and co-workers investigated phthalocyanine–rhodamine conjugates synthesized bonds, through amide bond formation, between tetrasulphosubstituted zinc phthalocyanine (ZnPcS_4_) and rhodamine dyes (either rhodamine 6G or rhodamine B), resulting in structures bearing four and approximately two rhodamine moieties per ZnPc, respectively (Figure 18, **XLVI** and **XLVII**) [105]. These conjugates, while not water-soluble due to the cationic nature of the rhodamine groups, were studied in DMSO, where photophysical analyses revealed a broadened and red-shifted Q-band absorption compared to native ZnPcS_4_ (684 nm for Pc; 539 nm and 564 nm for rhodamine 6G and B, in DMSO, respectively). The red shift and broadening are attributed to a loss of molecular symmetry upon conjugation. Spectroscopic measurements confirmed efficient intramolecular energy transfer from the rhodamine units to the phthalocyanine core, a key photophysical feature of the system. However, this conjugation also led to a roughly twofold decrease in fluorescence, singlet oxygen generation, and photodegradation quantum yields. This decline is likely due to intramolecular photoinduced charge transfer processes between the rhodamine and ZnPcS_4_ components, promoting non-radiative deactivation of the excited state. These results underscore the complexity of designing multichromophoric systems, where the intended energy transfer can be counterbalanced by competing charge transfer pathways that diminish the photosensitizing efficiency [105].

In other studies, Muli et al. reported the synthesis and biological evaluation of two mitochondria-targeting asymmetric zinc phthalocyanine–rhodamine B (ZnPc–RhB) conjugates (Figure 18, **XLVIII** and **XLIX**), designed to combine the photodynamic efficiency of phthalocyanines with the organelle-targeting and aggregation-reducing properties of rhodamine B [106]. The conjugates, **XLVIII** and **XLIX**, were obtained via DIC-mediated esterification of asymmetric ZnPc–OH precursors with RhB, yielding near-infrared (NIR)-absorbing photosensitizers with Q bands in DMSO at 682 and 713 nm (ZnPc) and 566 nm (RhB).

Both conjugates demonstrated the ability to generate singlet oxygen in solvents like DMSO, THF, and methanol, although their Φ_Δ_ values were somewhat lower than those of the non-conjugated analogs. Importantly, fluorescence aggregation studies showed that Rh B conjugation reduced aggregation in aqueous PBS, enhancing solubility and potentially contributing to higher photodynamic efficiency. Conjugates also exhibited increased hydrophobicity, aiding cellular membrane permeation. Biologically, both conjugates displayed negligible dark toxicity up to 15 µM concentrations, but under light irradiation (700 ± 40 nm, 45 J/cm^2^), they induced 70% cell death in colon 26 cells. In contrast, the unconjugated ZnPc derivatives showed no phototoxicity under identical conditions. These results demonstrate that RhB not only enhances mitochondrial localization due to its delocalized lipophilic cationic (DLC) nature but also significantly boosts the photodynamic performance of ZnPc through reduced aggregation and improved cell uptake, despite slightly reduced singlet oxygen generation [106].

A novel phthalocyanine–fluorescein conjugate (Figure 18, **L**) was synthesized by Ün et al. as a promising theranostic agent for PDT, enabling simultaneous imaging and treatment capabilities [107]. The ZnPc core was functionalized with three (dimethyl-1,3-dioxolan-4-yl)methoxy groups to enhance solubility and photostability and was subsequently conjugated to fluorescein through a PEG linker using a Cu(I)-catalyzed azide–alkyne cycloaddition. The resulting compound exhibited a strong Q-band absorption at 699 nm in DMSO, characteristic of effective NIR photosensitizers, and a fluorescence quantum yield of 0.11, which was lower than that of the unmodified ZnPc, likely due to intramolecular quenching or energy transfer between the two chromophores. A structurally analogous control compound lacking the fluorescein moiety but retaining the triazole linker was also synthesized to isolate the effect of fluorescein on the conjugate’s photophysical behavior. This comparison confirmed that the observed optical differences were attributable to the fluorescein unit. Although the conjugate demonstrated reduced fluorescence efficiency, its dual optical features and favorable synthetic accessibility (83% yield) suggest strong potential for use in PDT applications where both imaging and phototoxicity are desired [107].

## 7. Other Bioconjugates and Multifunctional Agents

### 7.1. Chalcones

Chalcones are a class of naturally occurring and synthetically accessible compounds characterized by an open-chain flavonoid structure consisting of two aromatic rings linked by a three-carbon α,β-unsaturated carbonyl system [108]. They exhibit a wide range of biological activities, including anti-inflammatory, antioxidant, antimicrobial, anticancer, and antiviral effects [109]. Their structural simplicity, ease of derivatization, and ability to interact with diverse biological targets make chalcones valuable scaffolds in medicinal chemistry [110]. Through modulation of key cellular pathways such as NF-κB, MAPK, and apoptosis signaling, chalcones have shown promise in the development of therapeutic agents for chronic diseases and cancer, including photosensitizers for PDT. Aiming to address this goal, in 2012, Tuncel et al. synthesized and characterized two types of conjugates: an unsymmetrical one, developed through AB_3_-type regioselective condensation, linking a chalcone moiety to the phthalocyanine core via a tetraethylene glycol (PEG-like) spacer to enhance water solubility and amphiphilicity [111], and its symmetrical derivative— a tetrahydroxylated ZnPc-chalcone conjugate (Figure 19, **LI**) [112]. Both were designed as dual-function agents for PDT and antiangiogenic intervention (vascular disrupting agents) in cancer treatment. The conjugates were formed via a carbamate bond between the phthalocyanine core and a chalcone moiety. The synthetic route utilized an isocyanate derivative of aminochalcone, yielding the final product in 74% yield [112]. Spectroscopic analysis showed that the Q-band absorption (704 nm in acetonitrile) of the ZnPc core was unaffected by conjugation, retaining its optical properties critical for PDT efficacy. Despite a drop in singlet oxygen generation efficiency in the case of the unsymmetrical derivative (Φ_Δ_ = 0.55 vs. 0.83 for unmodified ZnPc), the conjugate exhibited enhanced in vitro phototoxicity against HT-29 colon cancer cells, with an IC_50_ of 10.8 µM (red light λ > 600 nm), likely due to improved cellular uptake and selective accumulation. The conjugate also retained modest anti-migratory and cytotoxic effects on endothelial cells, aligning with the biological profile of free chalcone. However, the close proximity of the chalcone to the ZnPc was found to partially quench singlet oxygen production [111]. In the symmetrical conjugate, the chalcone moiety significantly increased the overall hydrophobicity of the molecule, overshadowing the intended solubilizing effect of the PEG linker. The conjugate showed exclusive partitioning into the octanol phase, confirming poor aqueous solubility, which precluded immediate biological testing [112].

Other chalcone derivatives were conjugated to ZnPc by Aribi et al. [113]. Three distinct conjugates were prepared, differing in the nature of the linker between the phthalocyanine and chalcone moieties: a non-cleavable ether linkage and two cleavable spacers, triazole-amide and triazole-carbamate, intended to allow for stimuli-responsive release of the chalcone in the tumor microenvironment (Figure 19, **LII**). This study provided a foundation for controlled-release PDT agents, where the release of the antivascular component (chalcone) could be selectively triggered by tumor-specific stimuli (e.g., low pH, enzymatic activity). The synthesis employed classical nucleophilic substitution and copper(I)-catalyzed azide–alkyne cycloaddition, with optimized yields (up to 72%) and structural verification by high-resolution mass spectrometry [113]. Spectroscopic analysis showed that the conjugation strategy had no adverse effect on the UV–Vis absorption features of the Pc core, with Q-bands consistently observed at ~703–704 nm (DMSO), indicating retained photophysical integrity and presumed capacity for singlet oxygen generation, essential for photodynamic therapy. Although no biological tests were reported in this study, the authors emphasized the importance of linker cleavability to optimize chalcone release, referencing prior observations that insufficient release led to diminished antivascular effects in earlier constructs [113].

In 2020, Ha et al. [114] developed a sophisticated multifunctional phthalocyanine-based therapeutic conjugate for targeted and controllable dual photodynamic and chemotherapy. The construct consists of a ZnPc core modified with three distinct functional modules: a 2,4-dinitrobenzenesulfonate (DNBS) moiety that quenches photoactivity and fluorescence until removed by intracellular glutathione (GSH); a chemo-prodrug based on combretastatin A-4 (CA4) tethered via a singlet oxygen-cleavable aminoacrylate linker; and a biotin ligand for tumor-targeted cellular uptake via biotin-receptor-mediated endocytosis (Figure 19, **LIII**) [114]. The synthesis employed a copper-catalyzed azide–alkyne cycloaddition (CuAAC) strategy to assemble the modular architecture with high specificity and yield. Spectroscopic characterization revealed a Q-band at 675 nm in DMF, with the DNBS group effectively quenching both fluorescence (Φ_F_ = 0.01) and singlet oxygen generation (Φ_Δ_ = 0.01). Upon intracellular exposure to GSH, DNBS was cleaved, restoring ZnPc photoactivity (Φ_Δ_ = 0.56 in DMF) and enabling both imaging and ROS generation. Simultaneously, the generated singlet oxygen cleaves the aminoacrylate bond, releasing active CA4, a microtubule destabilizer, in a spatiotemporally controlled manner. Biological validation showed preferential uptake by HepG2 liver cancer cells (which overexpress biotin receptors), with up to 8-fold higher fluorescence compared to HCT-116 cells. Moreover, GSH-activated cells exhibited 8–9-fold greater fluorescence than controls treated with GSH inhibitors, demonstrating the conjugate’s dual-level selectivity (GSH- and receptor-based). Cytotoxicity assays confirmed that PDT and chemotherapeutic effects worked synergistically, with a combination index (CI) < 1, indicating improved efficacy over individual treatments [114].

### 7.2. Other Compounds

Several other biologically active compounds have been used for the synthesis of Pc-based conjugates, including some nucleosides and polyamines: spermine, a naturally occurring polyamine involved in cellular growth and differentiation, known for its ability to interact with nucleic acids and stabilize DNA structure, and tetraazaadamantane, a nitrogen-rich polyamine structure derived from adamantane, known for its high stability and potential to form strong coordination complexes, making it useful in drug delivery, catalysis, and molecular recognition applications.

Regarding the first group, Reddy and co-workers reported the design and synthesis of novel trifluoroethoxy-substituted ZnPc covalently linked to deoxyribonucleosides, specifically ethynyluridine and ethynyladenosine, through twofold Sonogashira cross-coupling reactions. The resulting conjugates, Zn-CF_3_-Pc-U (**LIV**) and Zn-CF_3_-Pc-Ad (**LV**), were obtained in good yields and featured 12 peripheral trifluoroethoxy groups on the phthalocyanine macrocycle (Figure 20) [115].

These fluorinated phthalocyanines demonstrated distinct photophysical behavior compared to conventional phthalocyanine derivatives. Notably, unlike previously studied tert-butylated cytidine–phthalocyanine hybrids that tend to aggregate [116], the fluorinated conjugates favored monomeric states, enhancing their solubility and stability. Their photosensitivity was also found to be tunable depending on the solvent environment or the presence of a base, illustrating how peripheral fluorination can finely modulate photochemical properties. The incorporation of perfluoroalkyl chains significantly influenced the physicochemical profile of the conjugates, expanding the known benefits of fluorine substitution in phthalocyanine chemistry. Furthermore, the short molecular spacer between the nucleoside and the phthalocyanine core may allow for precise DNA strand scission, potentially enhancing site-selective photodynamic DNA damage [115].

In another study, Das and colleagues developed a series of ZnPc conjugates bearing perfluoroisopropyl substituents covalently linked to deoxyribonucleosides—specifically, deoxyadenosine and deoxyuridine—via Sonogashira cross-coupling [117]. The introduction of perfluoroisopropyl groups onto the phthalocyanine periphery was found to significantly improve key therapeutic properties. The resulting conjugates exhibited strong absorption at longer wavelengths, indicative of molecular asymmetry, and maintained monomeric behavior in both organic and aqueous media. Fluorescence quantum yields were high across different solvents, with slightly enhanced values observed in chloroform for conjugates containing TBDMS-protected nucleosides. Moreover, the compounds showed excellent chemical and photochemical stability regardless of the medium. Biological assessments demonstrated potent photocytotoxicity. In particular, three out of four conjugates achieved over 80% cell death in B16-F10 melanoma cells at a concentration of 50 μg/mL upon 664 nm laser irradiation. Additionally, these conjugates were effective against HT-1080 fibrosarcoma cells, highlighting their broad cytotoxic potential under photodynamic conditions [117].

In 2015, Ogbodu and colleagues reported an enhanced photodynamic therapeutic efficacy of zinc mono carboxy phenoxy phthalocyanine (ZnMCPPc) through conjugation with the polyamine spermine, forming ZnMCPPc-spermine, and further functionalization onto SWCNTs, yielding ZnMCPPc-spermine-SWCNT hybrid material. These modifications aimed to improve the photosensitizer’s cellular uptake, photophysical behavior, and targeting capability against MCF-7 breast cancer cells. The conjugation of spermine to ZnMCPPc was achieved via amide bond formation using DCC/DMAP chemistry [118]. Importantly, neither the conjugation with spermine nor the adsorption onto carbon nanotubes significantly altered the fluorescence quantum yield of the parent ZnMCPPc. However, the conjugate and the hybrid material demonstrated substantial increases in triplet state formation and singlet oxygen quantum yields, 0.83 for the conjugate and 0.62 for the material, compared to only 0.20 for the unmodified ZnMCPPc, all in DMSO solution. This enhancement was primarily attributed to the structural design incorporating a phenyl bridge, which facilitates a spin–orbit charge-transfer intersystem crossing (SOCT-ISC) mechanism, especially prominent in the conjugate. In vitro studies on MCF-7 cells revealed that all complexes were non-toxic under dark conditions, even at concentrations as high as 40 μM. Upon electric quartz lamp light activation (λ ≈600–1000 nm), however, a marked phototoxic response was observed. Parent Pc achieved a 64% reduction in cell viability at 40 μM, whereas the spermine-conjugated complex and the SWCNT-bound complex reduced cell viability by 97% and 95%, respectively, under identical conditions [118].

Regarding cyclic polyamine–tetraazaadamantane, a new type of phthalocyanine containing 3,5,7-trimethyl-1,4,6,10-tetraazatricyclo[3.3.1.13,7]decane-4,6,10-triol (1,4,6,10-tetraazaadamantane) has been synthesized by Tolbin et al. [119]. Its spectral properties and interaction with human lymphoblastoma cells infected with the HIV-1BRU strain have been studied. It has been shown to manifest the maximum anti-HIV activity when introduced simultaneously with the virus. It has been found that the highest activity of conjugate was reached at a concentration of 7.1 × 10^–6^ mol dm^–3^ [119].

## 8. Drawbacks and Limitations

In the development of phthalocyanine-based conjugates with various biomolecules, including therapeutic agents, targeting moieties such as biotin and folic acid, and photoactive chromophores like BODIPY and porphyrins, researchers have encountered several significant limitations. These challenges have a profound impact on the physicochemical, photophysical, and biological performance of the resulting conjugates. One of the most commonly observed drawbacks is the pronounced tendency of such conjugates to undergo aggregation, particularly when both components possess π-conjugated systems [34]. This π–π stacking can severely compromise solubility, limit bioavailability, and diminish the efficacy of ROS generation. To address this, strategies such as the incorporation of sterically bulky substituents, hydrophilic spacers, or dendritic architectures around the phthalocyanine core have been proposed to sterically hinder aggregation and improve dispersion in biological environments.

Another prominent issue is the decrease in aqueous solubility of the conjugates [41], which poses a barrier to clinical translation due to administration challenges. The use of hydrophilic linkers (e.g., polyethylene glycol chains of optimal length) or conjugation to solubility-enhancing carriers like cyclodextrins and polymers (e.g., polyvinylpyrrolidone or hyaluronic acid) can mitigate this problem by enhancing water compatibility while preserving photodynamic activity. Synthetic challenges further limit the widespread application of these conjugates. Many systems exhibit low to moderate yields (<20%) during the key cyclotetramerization step needed to form the phthalocyanine macrocycle [35]. Additionally, degradation of the phthalocyanine core during post-synthetic modification [103] and the formation of side products [79] remain persistent obstacles. Employing milder reaction conditions, protecting groups during functionalization, or pre-functionalizing the phthalonitrile precursors may improve yields and reduce side reactions. Optimizing catalyst systems and reaction solvents tailored for each specific conjugate can also significantly enhance synthetic efficiency. When employing ethylene glycol-based linkers, especially longer chains, reduced singlet oxygen generation and phototoxic efficacy have been reported [35]. This may result from increased conformational flexibility leading to inefficient energy transfer. Rational linker design, such as using rigid or semi-rigid linkers, or introducing branching points, can preserve spatial alignment for efficient energy and electron transfer. In systems utilizing ROS- or pH-cleavable linkers to achieve controlled release of the biomolecule, it is crucial to fine-tune the cleavability to ensure optimal activation within the tumor microenvironment or intracellular compartments [113]. Employing linkers with precisely calibrated cleavage thresholds and conducting in vitro/in vivo correlation studies can aid in achieving consistent bioavailability.

A further complication arises from the biomolecule itself interfering with ROS production, typically through competitive electron or energy transfer pathways [105]. This phenomenon is particularly evident in conjugates incorporating other photoactive molecules (e.g., BODIPY, porphyrins, rhodamine B), which may act as energy sinks, thereby lowering ROS yields. To overcome this, careful spectral and electronic matching of donor-acceptor pairs is necessary. Employing orthogonal photoactivation strategies or energy transfer insulation via spatial separation (e.g., rigid linkers) may preserve photodynamic efficacy. In some cases, encapsulation of the phthalocyanine conjugates in nanocarriers has led to a reduction in ROS generation compared to the free conjugate [64]. This suggests that encapsulation should be carefully optimized to avoid shielding the photosensitizer from light or oxygen. Surface-functionalized nanoparticles or responsive release systems that liberate the active conjugate under tumor-specific stimuli may offer a viable solution.

Finally, conjugation to biomolecules has also been shown to quench the native fluorescence of phthalocyanine, likely due to intramolecular interactions and energy transfer mechanisms [44]. Strategies to counteract this include tuning the spatial orientation between the phthalocyanine and biomolecule, using Förster resonance energy transfer (FRET)-resistant linkers, or employing time-gated fluorescence detection techniques to distinguish quenched states.

## 9. Summary and Conclusions

Phthalocyanine-based conjugates have become highly versatile tools for advancing PDT, uniting light-activated reactive oxygen species generation with targeted drug delivery and controlled activation. By covalently linking zinc phthalocyanine to chemotherapeutic agents such as erlotinib, lenvatinib, oxaliplatin, doxorubicin, tamoxifen, and camptothecin, researchers have created dual-action systems that leverage receptor-targeted uptake, enzyme-cleavable linkers, and synergistic cytotoxicity while minimizing off-target effects. These designs often include pH-responsive nanoparticles and multi-ligand systems capable of overcoming multidrug resistance, demonstrating improved tumor selectivity and reduced systemic toxicity. Beyond cancer drugs, conjugates with antibiotics like ciprofloxacin and isoniazid have delivered light-triggered antimicrobial activity against resistant pathogens and biofilms, while antiparasitic agents such as artesunate enable sonodynamic–PDT combinations with remarkable ROS generation and tumor inhibition. Anti-inflammatory drugs (e.g., indomethacin), hormones (e.g., mestranol), and antidepressants (e.g., moclobemide) further extend PDT’s scope, offering receptor-specific targeting and stimuli-responsive delivery for diverse indications.

Targeting ligands such as folic acid, biotin, RGD peptides, EGF-binding peptides, carbohydrates, and amino acids have been widely employed to exploit overexpressed receptors on cancer cells, dramatically improving selective uptake, intracellular accumulation, and therapeutic efficacy. For example, folic acid and biotin conjugates show strong tumor-specific accumulation and enhanced light-induced cytotoxicity both in vitro and in vivo, while carbohydrate-conjugated ZnPcs leverage GLUT transporter expression for targeted delivery, even offering innovative approaches for conditions like psoriasis. Amino acid-based conjugates, including polylysine, oligolysine, and arginine derivatives, improve biocompatibility, water solubility, and cellular targeting, with successful applications in both antibacterial PDT and cancer therapy. Hybrid systems combining ZnPc–poly-L-lysine with metallic nanoparticles such as gold and silver further enhance antibacterial effects through synergistic mechanisms.

Additionally, incorporating fluorescent dyes and porphyrinoid structures has expanded the theranostic potential of Pc conjugates. BODIPY–ZnPc systems deliver extended near-infrared absorption, high singlet oxygen yields, and efficient energy transfer for improved light harvesting and imaging, with designs supporting pH-activated fluorescence and even molecular logic-gate behavior. Porphyrin–phthalocyanine dyads, methylpheophorbide a hybrids, and fused porphyrin–Pc architectures exhibit strong panchromatic absorption and efficient intramolecular energy transfer, enhancing light-harvesting capacity for advanced PDT applications. Meanwhile, 5-aminolevulinic acid conjugates combine ZnPc’s photosensitizing activity with intracellular biosynthesis of protoporphyrin IX, enabling dual-mode ROS generation for synergistic cytotoxicity. Other dye conjugates with rhodamine or fluorescein improve imaging, reduce aggregation, and enhance mitochondrial targeting, supporting diagnostic and therapeutic use.

Despite these advances, critical challenges remain. Aggregation due to π–π stacking reduces solubility and ROS generation, while poor aqueous solubility limits clinical translation. Synthetic complexity, including low yields, side-product formation, and degradation during functionalization, further complicates development. Flexible linkers can reduce energy transfer efficiency and phototoxicity, while cleavable linkers must be precisely tuned for activation in tumor microenvironments. Biomolecule conjugates may also interfere with ROS production through competing energy transfer pathways, especially when incorporating other photoactive moieties. Encapsulation in nanocarriers can shield photosensitizers from light or oxygen, necessitating careful design of responsive release systems. Finally, conjugation often quenches Pc fluorescence, challenging imaging applications. Addressing these issues through optimized molecular design, using hydrophilic or rigid linkers, protective groups, improved catalysts, and smart delivery systems will be essential for realizing the full clinical potential of Pc–biomolecule conjugates as precise, multifunctional, and effective PDT agents.

Overall, phthalocyanine–biomolecule conjugates represent a promising direction for developing highly selective, multifunctional photosensitizers for photodynamic therapy. By intelligently combining targeted delivery, controlled activation, and imaging capabilities, these systems can achieve greater therapeutic precision and reduced side effects. However, overcoming challenges like aggregation, limited solubility, synthetic complexity and ROS interference remains crucial. Continued innovation in molecular design and delivery strategies will be key to translating these advanced conjugates into effective, clinically viable treatments.

## Data Availability

Not applicable.

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
