# Peer review of "Phthalocyanines Conjugated with Small Biologically Active Compounds for the Advanced Photodynamic Therapy: A Review"

_molecules, 2025, doi:10.3390/molecules30153297_

Round 1
Reviewer 1 Report
Comments and Suggestions for Authors
This is a Review paper that presents the synthetic knowledge about synthesis of phthalocyanine linked to biologically - active molecules.
Critical comments:
1. The main concern is that the theme is not fully covered. The authors may prepare several papers. In such case the study will be completed all existing knowledge on the topic for each class bioactive molecules with literatures incl. the newest papers.
2. My suggestion is to divide the biomolecules in some rules - for example natural, synthetic, drugs antibacterial, anticancer, or small biomolecules and macromolecules.
3. Many parts in the text are without appropriate citations or lack of citations.
Author Response
Dear Reviewer,
We have addressed all your critical comments and improved the manuscript. We have uploaded our response as a separate PDF file (please see the attachment).
Kind regards,
Tomasz Koczorowski, PhD

Reviewer 2 Report
Comments and Suggestions for Authors
I reviewed the manuscript entitled: Phthalocyanines conjugated with biologically active compounds for the advanced photodynamic therapy - a review (Manuscript ID: molecules-3787169) submitted to Molecules. This review paper describes the synthesis procedures of phthalocyanines and preparation of their conjugates with biologically active materials. Their photochemical properties are also introduced. The following issues should be considered before the publication.
1) This paper is the review article of phthalocyanine. I would like to recommend the addition of topics about photoimmunotherapy using phthalocyanine photosensitizer.
2) “RGD and EGF peptides” means “Arginylglycylaspartic acid” and “Epidermal Growth Factor”? If so, I would like to recommend the use of “Arginylglycylaspartic acid (RGD) and Epidermal Growth Factor (EGF) peptides”.
3) Several abbreviation should be opened at the first appearance for readers.
4) Fluorescence quantum yields and singlet oxygen quantum yields are presented in the text (for example. lines 214 and 215). Fluorescence lifetimes were also provided. Because these values depend on the environment surrounding photosensitizer, the solvent must be added.
5) I do not feel that the figures excepting for the synthesis scheme is enough. Illustration to explain the activity is helpful for readers to understand the functions of photosensitizer. For example, line 178, pH-sensitive linkers, should be explained by use of figure.
6) About photocytotoxicity, IC50 (For example, Line 219): The wavelength information should be provided.
Author Response
Dear Reviewer,
We have addressed all your comments and improved the manuscript. We have uploaded our response as a separate PDF file (please see the attachment).
Kind regards,
Tomasz Koczorowski, PhD

Round 2
Reviewer 1 Report
Comments and Suggestions for Authors
The review was revised accordingly.
Reviewer 2 Report
Comments and Suggestions for Authors
I checked the reviewed manuscript (Manuscript ID: molecules-3787169). I feel that the issues have been resolved.